# Fault-Tolerant Federated Reinforcement Learning with Theoretical Guarantee

**Flint Xiaofeng Fan**[1, 3], **Yining Ma**[2], **Zhongxiang Dai**[1], **Wei Jing**[4],
**Cheston Tan**[3], **Bryan Kian Hsiang Low**[1]

[1]Dept. of Computer Science, National University of Singapore, Republic of Singapore
[2]Dept. of ISEM, National University of Singapore, Republic of Singapore
[3]Institute for Infocomm Research, A*STAR, Republic of Singapore
[4]Alibaba DAMO Academy, Hangzhou, China
[1]`{xiaofeng,daizhongxiang,lowkh}@comp.nus.edu.sg`,[2]`yiningma@u.nus.edu`
[3]`{stufanxf,cheston-tan}@i2r.a-star.edu.sg`,[4]`jw334405@alibaba-inc.com`

## Abstract

The growing literature of *Federated Learning* (FL) has recently inspired *Federated Reinforcement Learning* (FRL) to encourage multiple agents to federatively build a *better* decision-making policy without sharing raw trajectories. Despite its promising applications, existing works on FRL fail to I) provide theoretical analysis on its convergence, and II) account for random system failures and adversarial attacks. Towards this end, we propose the first FRL framework the convergence of which is guaranteed and tolerant to less than half of the participating agents being random system failures or adversarial attackers. We prove that the sample efficiency of the proposed framework is guaranteed to improve with the number of agents and is able to account for such potential failures or attacks. All theoretical results are empirically verified on various RL benchmark tasks.

## 1 Introduction

*Reinforcement learning* (RL) has recently been applied to many real-world decision-making problems such as gaming, robotics, healthcare, etc. [1–3]. However, despite its impressive performances in simulation, RL often suffers from poor sample efficiency, which hinders its success in real-world applications [4, 5]. For example, when RL is applied to provide clinical decision support [3, 6, 7], its performance is limited by the number (i.e., sample size) of admission records possessed by a hospital, which cannot be synthetically generated [3]. As this challenge is usually faced by many agents (e.g., different hospitals), a natural solution is to encourage multiple RL agents to share their trajectories, to collectively build a better decision-making policy that one single agent can not obtain by itself. However, in many applications, raw RL trajectories contain sensitive information (e.g., the medical records contain sensitive information about patients) and thus sharing them is prohibited. To this end, the recent success of *Federated Learning* (FL) [8–11] has inspired the setting of *Federated Reinforcement Learning* (FRL) [12], which aims to *federatively* build a *better* policy from multiple RL agents without requiring them to share their raw trajectories. FRL is practically appealing for addressing the sample inefficiency of RL in real systems, such as autonomous driving [13], fast personalization [14], optimal control of IoT devices [15], robots navigation [16], and resource management in networking [17]. Despite its promising applications, FRL is faced by a number of major challenges, which existing works are unable to tackle.

Firstly, existing FRL frameworks are not equipped with theoretical convergence guarantee, and thus lack an assurance for the sample efficiency of practical FRL applications, which is a critical drawback

35th Conference on Neural Information Processing Systems (NeurIPS 2021).

due to the high sampling cost of RL trajectories in real systems [4]. Unlike FL where training data can be collected offline, FRL requires every agent to sample trajectories by interacting with the environment during learning. However, interacting with real systems can be slow, expensive, or fragile. This makes it critical for FRL to be sample-efficient and hence highlights the requirement for convergence guarantee of FRL, without which no assurance on its sample efficiency is provided for practical applications. To fill this gap, we establish on recent endeavors in stochastic variance-reduced optimization techniques to develop a variance-reduced federated policy gradient framework, the convergence of which is guaranteed. We prove that the proposed framework enjoys a sample complexity of $O(1/\epsilon^{5/3})$ to converge to an $\epsilon$-stationary point in the single-agent setting, which matches recent results of variance-reduced policy gradient [18, 19]. More importantly, the aforementioned sample complexity is guaranteed to *improve* at a rate of $O(1/K^{2/3})$ upon the federation of $K$ agents. This guarantees that an agent achieves a better sample efficiency by joining the federation and benefits from more participating agents, which are highly desirable in FRL.

Another challenge inherited from FL is that FRL is vulnerable to random failures or adversarial attacks, which poses threats to many real-world RL systems. For example, robots may behave arbitrarily due to random hardware issues; clinical data may provide inaccurate records and hence create misleading trajectories [3]; autonomous vehicles, on which RL is commonly deployed, are subject to adversarial attacks [20]. As we will show in experiments, including such random failures or adversary agents in FRL can significantly deteriorate its convergence or even result in unlearnability. Of note, random failures and adversarial attacks in FL systems are being encompassed by the Byzantine failure model [21], which is considered as the most stringent fault formalism in distributed computing [22, 23] – a small fraction of agents may behave arbitrarily and possibly adversarially, with the goal of breaking or at least slowing down the convergence of the system. As algorithms proven to be correct in this setting are guaranteed to converge under arbitrary system behavior (e.g., exercising failures or being attacked) [9, 24], we study the fault tolerance of our proposed FRL framework using the Byzantine failure model. We design a gradient-based Byzantine filter on top of the variance-reduced federated policy gradient framework. We show that, when a certain percentage (denoted by $\alpha < 0.5$) of agents are Byzantine agents, the sample complexity of the FRL system is worsened by *only an additive term* of $O(\alpha^{4/3}/\epsilon^{5/3})$ (Section 4). Therefore, when $\alpha \to 0$, (i.e., an ideal system with zero chance of failure), the filter induces no impact on the convergence.

**Contributions.** In this paper, we study the *federated* reinforcement learning problem with theoretical guarantee in the potential presence of faulty agents. We introduce *Federated Policy Gradient with Byzantine Resilience* (FedPG-BR), the first FRL framework that is theoretically principled and practically effective for the FRL setting, accounting for random systematic failures and adversarial attacks. In particular, FedPG-BR (a) enjoys a guaranteed sample complexity which improves with more participating agents, and (b) is tolerant to the Byzantine fault in both theory and practice. We discuss the details of problem setting and the technical challenges (Section 3) and provide theoretical analysis of FedPG-BR (Section 4). We also demonstrate its empirical efficacy on various RL benchmark tasks (Section 5).

## 2 Background

**Stochastic Variance-Reduced Gradient** aims to solve $\min_{\boldsymbol{\theta} \in \mathbb{R}^d}[J(\boldsymbol{\theta}) \triangleq \frac{1}{B}\sum_{i=1}^{B} J_i(\boldsymbol{\theta})]$. Under the common assumption of all function components $J_i$ being smooth and convex in $\boldsymbol{\theta}$, *gradient descent* (GD) achieves linear convergence in the number of iterations of parameter updates [25, 26]. However, every iteration of GD requires $B$ gradient computations, which can be expensive for large $B$. To overcome this problem, *stochastic GD* (SGD) [27, 28] samples a single data point per iteration, which incurs lower per-iteration cost yet results in a sub-linear convergence rate [29]. For a better trade-off between convergence rate and per-iteration computational cost, the *stochastic variance-reduced gradient* (SVRG) method has been proposed, which reuses past gradient computations to reduce the variance of the current gradient estimate [30–33]. More recently, *stochastically controlled stochastic gradient* (SCSG) has been proposed for convex [34] or smooth non-convex objective function [35], to further reduce the computational cost of SVRG especially when required $\epsilon$ is small in finding $\epsilon$-approximate solution. Refer to Appendix A.1 for more details on SVRG and SCSG.

**Reinforcement Learning** (RL) can be modelled as a discrete-time Markov Decision Process (MDP) [36]: $M \triangleq \{\mathcal{S}, \mathcal{A}, \mathcal{P}, \mathcal{R}, \gamma, \rho\}$. $\mathcal{S}$ represents the state space, $\mathcal{A}$ is the action space, $\mathcal{P}(s'|s, a)$

defines the transition probability from state $s$ to $s'$ after taking action $a$, $\mathcal{R}(s,a) : \mathcal{S} \times \mathcal{A} \mapsto [0, R]$ is the reward function for state-action pair $(s, a)$ and some constant $R > 0$, $\gamma \in (0, 1)$ is the discount factor, and $\rho$ is the initial state distribution. An agent's behavior is controlled by a policy $\pi$, where $\pi(a|s)$ defines the probability that the agent chooses action $a$ at state $s$. We consider episodic MDPs with trajectory horizon $H$. A trajectory $\tau \triangleq \{s_0, a_0, s_1, a_1, ..., s_{H-1}, a_{H-1}\}$ is a sequence of state-action pairs traversed by an agent following any stationary policy, where $s_0 \sim \rho$. $\mathcal{R}(\tau) \triangleq \sum_{t=0}^{H-1} \gamma^t \mathcal{R}(s_t, a_t)$ gives the cumulative discounted reward for a trajectory $\tau$.

**Policy Gradient** (PG) methods have achieved impressive successes in model-free RL [37, 38, ,etc.]. Compared with deterministic value-function based methods such as Q-learning, PG methods are generally more effective in high-dimensional problems and enjoy the flexibility of stochasticity. In PG, we use $\pi_{\boldsymbol{\theta}}$ to denote the policy parameterized by $\boldsymbol{\theta} \in \mathbb{R}^d$ (e.g., a neural network), and $p(\tau|\pi_{\boldsymbol{\theta}})$ to represent the trajectory distribution induced by policy $\pi_{\boldsymbol{\theta}}$. For brevity, we use $\theta$ to denote the corresponding policy $\pi_{\boldsymbol{\theta}}$. The performance of a policy $\boldsymbol{\theta}$ can be measured by $J(\boldsymbol{\theta}) \triangleq \mathbb{E}_{\tau \sim p(\cdot|\boldsymbol{\theta})}[\mathcal{R}(\tau)|M]$. Taking the gradient of $J(\boldsymbol{\theta})$ with respect to $\boldsymbol{\theta}$ gives

$$\nabla_{\boldsymbol{\theta}} J(\boldsymbol{\theta}) = \int_{\tau} \mathcal{R}(\tau) \nabla_{\boldsymbol{\theta}} p(\tau \mid \boldsymbol{\theta}) \mathrm{d}\tau = \mathbb{E}_{\tau \sim p(\cdot|\boldsymbol{\theta})} \left[ \nabla_{\boldsymbol{\theta}} \log p(\tau \mid \boldsymbol{\theta}) \mathcal{R}(\tau) \mid M \right] \tag{1}$$

Then, the policy $\boldsymbol{\theta}$ can be optimized by gradient ascent. Since computing (1) is usually prohibitive, stochastic gradient ascent is typically used. In each iteration, we sample a batch of trajectories $\{\tau_i\}_{i=1}^B$ using the current policy $\boldsymbol{\theta}$, and update the policy by $\boldsymbol{\theta} \leftarrow \boldsymbol{\theta} + \eta \widehat{\nabla}_B J(\boldsymbol{\theta})$, where $\eta$ is the step size and $\widehat{\nabla}_B J(\boldsymbol{\theta})$ is an estimate of (1) using the sampled trajectories $\{\tau_i\}_{i=1}^B$: $\widehat{\nabla}_B J(\boldsymbol{\theta}) = \frac{1}{B} \sum_{i=1}^B \nabla_{\boldsymbol{\theta}} \log p(\tau_i \mid \boldsymbol{\theta}) \mathcal{R}(\tau_i)$. The most common policy gradient estimators, such as REINFORCE [39] and GPOMDP [40], can be expressed as

$$\widehat{\nabla}_B J(\boldsymbol{\theta}) = \frac{1}{B} \sum_{i=1}^B g(\tau_i | \boldsymbol{\theta}) \tag{2}$$

where $\tau_i = \{s_0^i, a_0^i, s_1^i, a_1^i, \ldots, s_{H-1}^i, a_{H-1}^i\}$ and $g(\tau_i|\boldsymbol{\theta})$ is an *unbiased* estimate of $\nabla_{\boldsymbol{\theta}} \log p(\tau_i \mid \boldsymbol{\theta}) \mathcal{R}(\tau_i)$. We provide formal definition of $g(\tau_i|\boldsymbol{\theta})$ in Appendix A.2.

**SVRPG.** A key issue for PG is the high variance of the estimator based on stochastic gradients (2) which results in slow convergence. Similar to SGD for finite-sum optimization, PG requires $O(1/\epsilon^2)$ trajectories to find an $\epsilon$-stationary point such that $\mathbb{E}[\|\nabla J(\boldsymbol{\theta})\|^2] \leq \epsilon$ [19]. That is, PG typically requires a large number of trajectories to find a well-performing policy. To reduce the variance of the gradient estimator in PG (2), SVRG has been applied to policy evaluation [41, 42] and policy optimization [43]. The work of Papini et al. [18] has adapted the theoretical analysis of SVRG to PG to introduce the *stochastic variance-reduced PG* (SVRPG) algorithm. More recently, Xu et al. [19] has refined the analysis of SVRPG [18] and shown that SVRPG enjoys a sample complexity of $O(1/\epsilon^{5/3})$. These works have demonstrated both theoretically and empirically that SVRG is a promising approach to reduce the variance and thus improve the sample efficiency of PG methods.

**Fault tolerance** refers to the property that enables a computing system to continue operating properly without interruption when one or more of its workers fail. Among the many fault formalisms, the Byzantine failure model has a rich history in distributed computing [22, 23] and is considered as the most stringent fault formalism in fault-tolerant FL system design [9, 24]. Originated from the *Byzantine generals problem* [21], the Byzantine failure model allows an $\alpha$-fraction (typical $\alpha < 0.5$) of workers to behave arbitrarily and possibly adversarially, with the goal of breaking or at least slowing down the convergence of the algorithm. As algorithms proven to be resilient to the Byzantine failures are guaranteed to converge under arbitrary system behavior (hence fault-tolerant) [22, 23], it has motivated a significant interest in providing distributed *supervised learning* with Byzantine resilience guarantees [e.g., 44–49]. However, there is yet no existing work studying the correctness of Byzantine resilience in the context of FRL.

## 3 Fault-tolerant federated reinforcement learning

### 3.1 Problem statement

Our problem setting is similar to that of FL [8] where a central server is assumed to be trustworthy and governs the federation of $K$ distributed agents $k \in \{1, ..., K\}$. In each round $t \in \{1, ..., T\}$,

the central server broadcasts its parameter $\boldsymbol{\theta}_0^t$ to all agents. Each agent then independently samples a batch of trajectories $\{\tau_{t,i}^{(k)}\}_{i=1}^{B_t}$ by interacting with the environment using the obtained policy, e.g., $\{\tau_{t,i}^{(k)}\}_{i=1}^{B_t} \sim p(\cdot|\boldsymbol{\theta}_0^t)$. However, different from FL where each agent computes the parameter updates and sends the updated parameter to the server for aggregation [8], agents in our setup do not compute the updates locally, but instead send the gradient computed w.r.t. their local trajectories $\mu_t^{(k)} \triangleq \widehat{\nabla}_{B_t} J(\boldsymbol{\theta}_0^t)$ directly to the server. The server then aggregates the gradients, performs a policy update step, and starts a new round of federation.

Of note, every agent including the server is operating in a separate copy of the MDP. No exchange of raw trajectories is required, and no communication between any two agents is allowed. To account for potential failures and attacks, we allow an $\alpha$-fraction of agents to be Byzantine agents with $\alpha \in [0, 0.5)$. That is, in each round $t$, a good agent always sends its computed $\mu_t^{(k)}$ back to the server, while a Byzantine agent may return any arbitrary vector.[1] The server has no information regarding whether Byzantine agents exist and cannot track the communication history with any agent. In every round, the server can only access the $K$ gradients received from agents, and thereby uses them to detect Byzantine agents so that it only aggregates the gradients from those agents that are believed to be non-Byzantine agents.

**Notations.** Following the notations of SCSG [35], we use $\boldsymbol{\theta}_0^t$ to denote the server's initial parameter in round $t$ and $\boldsymbol{\theta}_n^t$ to represent the updated parameter at the $n$-th step in round $t$. $\tau_{t,i}^{(k)}$ represents agent $k$'s $i$-th trajectory sampled using $\boldsymbol{\theta}_0^t$. $\|\cdot\|$ denotes Euclidean norm and Spectral norms for vectors and matrices, respectively. $O(\cdot)$ hides all constant terms.

### 3.2 Technical challenges

There is an emerging interest in Byzantine-resilient distributed *supervised learning* [e.g., 44–50]. However, a direct application of those works to FRL is not possible due to that the objective function $J(\boldsymbol{\theta})$ of RL, which is conditioned on $\tau \sim p(\cdot|\boldsymbol{\theta})$, is different from the supervised classification loss seen in the aforementioned works, resulting in the following issues:

*Non-stationarity*: unlike in supervised learning, the distribution of RL trajectories is affected by the value of the policy parameter which changes over time (e.g., $\tau \sim p(\cdot|\boldsymbol{\theta})$). We deal with the non-stationarity using importance sampling [51] (Section 3.3).

*Non-concavity*: the objective function $J(\boldsymbol{\theta})$ is typically non-concave. To derive the theoretical results accounting for the non-concavity, we need the $L$-smoothness assumption on $J(\boldsymbol{\theta})$, which is a reasonable assumption and commonly made in the literature [52] (Section 4). Hence we aim to find an $\epsilon$-approximate solution (i.e., a commonly used objective in non-convex optimization):

**Definition 1** ($\epsilon$-approximate solution). *A point $\boldsymbol{\theta}$ is called $\epsilon$-stationary if $\|\nabla J(\boldsymbol{\theta})\|^2 \leq \epsilon$. Moreover, the algorithm is said to achieve an $\epsilon$-approximate solution in $t$ rounds if $\mathbb{E}[\|\nabla J(\boldsymbol{\theta})\|^2] \leq \epsilon$, where the expectation is with respect to all randomness of the algorithm until round $t$.*

*High variance in gradient estimation:* the high variance in estimating (2) renders the FRL system vulnerable to variance-based attacks which conventional Byzantine-resilient optimization works fail to defend [47]. To combat this issue, we adapt the SCSG optimization [35] to federated policy gradient for a refined control over the estimation variance, hence enabling the following assumption which we exploit to design our Byzantine filtering step:

**Assumption 2** (On bounded variance of the gradient estimator). *There is a constant $\sigma$ such that $\|g(\tau|\boldsymbol{\theta}) - \nabla J(\boldsymbol{\theta})\| \leq \sigma$ for any $\tau \sim p(\tau|\boldsymbol{\theta})$ for all policy $\pi_{\boldsymbol{\theta}}$.*

**Remark.** *Assumption 2 is also seen in Byzantine-resilient optimization [46, 48] and may be relaxed to $\mathbb{E}\|g(\tau|\boldsymbol{\theta}) - \nabla J(\boldsymbol{\theta})\| \leq \boldsymbol{\theta}$ which is a standard assumption commonly used in stochastic non-convex optimization [e.g. 35, 53]. In this work, the value of $\sigma$ is the maximum difference between optimal gradient $\nabla J(\boldsymbol{\theta})$ and the gradient estimate $g(\tau|\boldsymbol{\theta})$ w.r.t. any trajectories induced by policy $\pi_{\boldsymbol{\theta}}$. For complex real-world problems with continuous, high-dimensional controls, $\sigma$ may be upper-bounded, provided that the MDP is Lipschitz continuousPirotta et al. [52]. The deviation can be obtained by referring to Proposition 2 of Pirotta et al. [52].[2]*

---

[1]A Byzantine agent may not be Byzantine in every round.

[2]The value of $\sigma$ can be estimated at the server.

### 3.3 Algorithm description

The pseudocode for the proposed *Federated Policy Gradient with Byzantine Resilience* (FedPG-BR) is shown in Algorithm 1. FedPG-BR starts with a randomly initialized parameter $\tilde{\boldsymbol{\theta}}_0$ at the server. At the beginning of the $t$-th round, the server keeps a snapshot of its parameter from the previous round (i.e., $\boldsymbol{\theta}_0^t \leftarrow \tilde{\boldsymbol{\theta}}_{t-1}$) and broadcasts this parameter to all agents (line 3). Every (good) agent $k$ samples $B_t$ trajectories $\{\tau_{t,i}^{(k)}\}_{i=1}^{B_t}$ using the policy $\boldsymbol{\theta}_0^t$ (line 5), computes a gradient estimate $\mu_t^{(k)} \triangleq 1/B_t \sum_{i=1}^{B_t} g(\tau_{t,i}^{(k)}|\boldsymbol{\theta}_0^t)$ where $g$ is either the REINFORCE or the GPOMDP estimator (line 6), and sends $\mu_t^{(k)}$ back to the server. For a Byzantine agent, it can send an arbitrary vector instead of the correct gradient estimate. After all gradients are received, the server performs the *Byzantine filtering* step, and then computes the batch gradient $\mu_t$ by averaging those gradients that the server believes are from non-Byzantine agents (line 7). For better clarity, we present the subroutine **FedPG-Aggregate** for Byzantine filtering and gradient aggregation in Algorithm 1.1, which we discuss in detail separately.

The aggregation is then followed by the SCSG inner loop [35] with $N_t$ steps, where $N_t$ is sampled from a geometric distribution with parameter $\frac{B_t}{B_t+b_t}$ (line 8). At step $n$, the server *independently* samples $b_t$ ($b_t \ll B_t$) trajectories $\{\tau_{n,j}^t\}_{j=1}^{b_t}$ using its current policy $\boldsymbol{\theta}_n^t$ (line 10), and then updates the policy parameter $\boldsymbol{\theta}_n^t$ based on the following semi-stochastic gradient (lines 11 and 12):

$$v_n^t \triangleq \frac{1}{b_t} \sum_{j=1}^{b_t} \left[ g(\tau_{n,j}^t|\boldsymbol{\theta}_n^t) - \omega(\tau_{n,j}^t|\boldsymbol{\theta}_n^t,\boldsymbol{\theta}_0^t)g(\tau_{n,j}^t|\boldsymbol{\theta}_0^t) \right] + \mu_t. \tag{3}$$

The last two terms serve as a correction to the gradient estimate to reduce variance and improve the convergence rate of Algorithm 1. Of note, the semi-stochastic gradient above (3) differs from that used in SCSG due to the additional term of $\omega(\tau|\boldsymbol{\theta}_n^t,\boldsymbol{\theta}_0^t) \triangleq p(\tau|\boldsymbol{\theta}_0^t)/p(\tau|\boldsymbol{\theta}_n^t)$. This term is known as the *importance weight* from $p(\tau|\boldsymbol{\theta}_n^t)$ to $p(\tau|\boldsymbol{\theta}_0^t)$ to account for the aforementioned non-stationarity of the distribution in RL [18, 19]. In particular, directly computing $g(\tau_{n,j}^t|\boldsymbol{\theta}_0^t)$ results in a biased estimation because the trajectories $\{\tau_{n,j}^t\}_{j=1}^{b_t}$ are sampled from the policy $\boldsymbol{\theta}_n^t$ instead of $\boldsymbol{\theta}_0^t$. We prove in Lemma 8 (Appendix E) that this importance weight results in an unbiased estimate of the gradient, i.e., $\mathbb{E}_{\tau \sim p(\cdot|\boldsymbol{\theta}_n)}[\omega(\tau|\boldsymbol{\theta}_n,\boldsymbol{\theta}_0)g(\tau|\boldsymbol{\theta}_0)] = \nabla J(\boldsymbol{\theta}_0)$.

Here we describe the details of our Byzantine filtering step (i.e., the subroutine **FedPG-Aggregate** in Algorithm 1.1), which is inspired by the works of Alistarh et al. [46] and Khanduri et al. [48] in distributed supervised learning. In any round $t$, we use $\mathcal{G}$ to denote the set of true good agents and use $\mathcal{G}_t$ to denote the set of agents that are believed to be good by the server. Our Byzantine filtering

---

**Algorithm 1** FedPG-BR

1: **Input:** $\tilde{\boldsymbol{\theta}}_0 \in \mathbb{R}^d$, batch size $B_t$, mini batch size $b_t$, step size $\eta_t$
2: **for** $t = 1$ **to** $T$ **do**
3:     $\boldsymbol{\theta}_0^t \leftarrow \tilde{\boldsymbol{\theta}}_{t-1}$                                                *; broadcast to all agents*
4:     **for** $k = 1$ **to** $K$ **do**
5:         Sample $B_t$ trajectories $\{\tau_{t,i}^{(k)}\}_{i=1}^{B_t}$ from $p(\cdot|\boldsymbol{\theta}_0^t)$
6:         $\mu_t^{(k)} \triangleq \begin{cases} \frac{1}{B_t} \sum_{i=1}^{B_t} g(\tau_{t,i}^{(k)}|\boldsymbol{\theta}_0^t) & \text{for } k \in \mathcal{G} \\ * & \text{for } k \notin \mathcal{G} \end{cases}$        *; push $\mu_t^{(k)}$ to server*
7:     $\mu_t \leftarrow$ **FedPG-Aggregate**$(\{\mu_t^{(k)}\}_{k=1}^K)$
8:     Sample $N_t \sim Geom(\frac{B_t}{B_t+b_t})$
9:     **for** $n = 0$ **to** $N_t - 1$ **do**
10:        Sample $b_t$ trajectories $\{\tau_{n,j}^t\}_{j=1}^{b_t}$ from $p(\cdot|\boldsymbol{\theta}_n^t)$
11:        $v_n^t \triangleq \frac{1}{b_t} \sum_{j=1}^{b_t} [g(\tau_{n,j}^t|\boldsymbol{\theta}_n^t) - \omega(\tau_{n,j}^t|\boldsymbol{\theta}_n^t,\boldsymbol{\theta}_0^t)g(\tau_{n,j}^t|\boldsymbol{\theta}_0^t)] + \mu_t$
12:        $\boldsymbol{\theta}_{n+1}^t = \boldsymbol{\theta}_n^t + \eta_t v_n^t$
13:     $\tilde{\boldsymbol{\theta}}_t \leftarrow \boldsymbol{\theta}_{N_t}^t$
14: **Output:** $\tilde{\boldsymbol{\theta}}_a$ uniformly randomly picked from $\{\tilde{\boldsymbol{\theta}}_t\}_{t=1}^T$

---

**Algorithm 1.1 FedPG-Aggregate**

---

1: **Input:** Gradient estimates from $K$ agents in round $t$: $\{\mu_t^{(k)}\}_{k=1}^K$, variance bound $\sigma$, filtering threshold $\mathfrak{T}_\mu \triangleq 2\sigma\sqrt{\frac{V}{B_t}}$, where $V \triangleq 2\log(\frac{2K}{\delta})$ and $\delta \in (0,1)$

2: $S_1 \triangleq \{\mu_t^{(k)}\}$ where $k \in [K]$ s.t. $\left|\left\{k' \in [K] : \left\|\mu_t^{(k')} - \mu_t^{(k)}\right\| \leq \mathfrak{T}_\mu\right\}\right| > \frac{K}{2}$

3: $\mu_t^{\text{mom}} \leftarrow \text{argmin}_{\mu_t^{(\tilde{k})}} \|\mu_t^{(\tilde{k})} - \text{mean}(S_1)\|$ where $\tilde{k} \in S_1$

4: *R1:* $\mathcal{G}_t \triangleq \left\{k \in [K] : \left\|\mu_t^{(k)} - \mu_t^{\text{mom}}\right\| \leq \mathfrak{T}_\mu\right\}$

5: **if** $|\mathcal{G}_t| < (1-\alpha)K$    **then**

6:    $S_2 \triangleq \{\mu_t^{(k)}\}$ where $k \in [K]$ s.t. $\left|\left\{k' \in [K] : \left\|\mu_t^{(k')} - \mu_t^{(k)}\right\| \leq 2\sigma\right\}\right| > \frac{K}{2}$

7:    $\mu_t^{\text{mom}} \leftarrow \text{argmin}_{\mu_t^{(\tilde{k})}} \|\mu_t^{(\tilde{k})} - \text{mean}(S_2)\|$ where $\tilde{k} \in S_2$

8:    *R2:* $\mathcal{G}_t \triangleq \left\{k \in [K] : \left\|\mu_t^{(k)} - \mu_t^{\text{mom}}\right\| \leq 2\sigma\right\}$

9: **Return:** $\mu_t \triangleq \frac{1}{|\mathcal{G}_t|}\sum_{k \in \mathcal{G}_t}\mu_t^{(k)}$

---

consists of two filtering rules denoted by R1 (lines 2-4) and R2 (lines 6-8). R2 is more intuitive to understand, so we start by introducing R2. Firstly, in line 6, the server constructs a set $S_2$ of *vector medians* [46] where each element of $S_2$ is chosen from $\{\mu_t^{(k)}\}_{k=1}^K$ if it is close (within $2\sigma$ in Euclidean distance) to more than $K/2$ elements. Next, the server finds a *Mean of Median* vector $\mu_t^{\text{mom}}$ from $S_2$, which is defined as any $\mu_t^{(\tilde{k})} \in S_2$ that is the closet to the mean of the vectors in $S_2$. After $\mu_t^{\text{mom}}$ is selected, the server can construct the set $\mathcal{G}_t$ by filtering out any $\mu_t^{(k)}$ whose distance to $\mu_t^{\text{mom}}$ is larger than $2\sigma$ (line 8). This filtering rule is designed based on Assumption 2 which implies that the maximum distance between any two good agents is $2\sigma$, and our assumption that at least half of the agents are good (i.e., $\alpha < 0.5$). We show in Appendix D that under these two assumptions, R2 guarantees that all good agents are included in $\mathcal{G}_t$ (i.e., $|\mathcal{G}_t| \geq (1-\alpha)K$). We provide a graphical illustration (Fig. 4 in Appendix D) on that if any Byzantine agent is included in $\mathcal{G}_t$, its distance to the true gradient $\nabla J(\boldsymbol{\theta}_0^t)$ is at most $3\sigma$, which ensures that its impact on the algorithm is limited. Note that the pairwise computation among the weights of all the agents can be implemented using the Euclidean Distance Matrix Trick [54].

R1 (lines 2-4) is designed in a similar way: R1 ensures that all good agents are *highly likely* to be included in $\mathcal{G}_t$ by exploiting Lemma 14 (Appendix E) to guarantee that *with high probability*, all good agents are *concentrated in a smaller region*. That is, define $V \triangleq 2\log(2K/\delta)$ and $\delta \in (0,1)$, then with probability of $\geq 1 - \delta$, the maximum distance between any two good agents is $\mathfrak{T}_\mu \triangleq 2\sigma\sqrt{V/B_t}$. Having all good agents in a smaller region improves the filtering strategy, because it makes the Byzantine agents less likely to be selected and reduces their impact even if they are selected. Therefore, R1 is applied first such that if R1 fails to include all good agents in $\mathcal{G}_t$ (line 4) which happens with probability $< \delta$, R2 is then employed as a backup to ensure that $\mathcal{G}_t$ always include all good agents. Therefore, these two filtering rules ensure in any round $t$ that (a) gradients from good agents are never filtered out, and that (b) if gradients from Byzantine agents are not filtered out, their impact is limited since their maximum distance to $\nabla J(\boldsymbol{\theta}_0^t)$ is bounded by $3\sigma$.

## 4 Theoretical results

Here, we firstly put in place a few assumptions required for our theoretical analysis, all of which are common in the literature.

**Assumption 3** (On policy derivatives). *Let $\pi_{\boldsymbol{\theta}}(a|s)$ be the policy of an agent at state $s$. There exist constants $G, M > 0$ s.t. the log-density of the policy function satisfies, for all $a \in \mathcal{A}$ and $s \in \mathcal{S}$*

$$|\nabla_{\boldsymbol{\theta}} \log \pi_{\boldsymbol{\theta}}(a|s)| \leq G, \qquad \|\nabla_{\boldsymbol{\theta}}^2 \log \pi_{\boldsymbol{\theta}}(a|s)\| \leq M, \qquad \forall a \in \mathcal{A}, \forall s \in \mathcal{S}$$

Assumption 3 provides the basis for the smoothness assumption on the objective function $J(\boldsymbol{\theta})$ commonly used in non-convex optimization [32, 33] and also appears in Papini et al. [18], Xu et al. [19]. Specifically, Assumption 3 implies:

**Proposition 4** (On function smoothness). *Under Assumption 3, $J(\boldsymbol{\theta})$ is L-smooth with $L \triangleq HR(M + HG^2)/(1 - \gamma)$. Let $g(\tau|\boldsymbol{\theta})$ be the REINFORCE or GPOMDP gradient estimators. Then for all $\boldsymbol{\theta}, \boldsymbol{\theta}_1, \boldsymbol{\theta}_2 \in \mathbb{R}^d$, it holds that*

$$\|g(\tau|\boldsymbol{\theta})\| \leq C_g, \qquad \|g(\tau \mid \boldsymbol{\theta}_1) - g(\tau \mid \boldsymbol{\theta}_2)\| \leq L_g \|\boldsymbol{\theta}_1 - \boldsymbol{\theta}_2\|$$

*where $L_g \triangleq HM(R + |C_b|)/(1 - \gamma), C_g \triangleq HG(R + |C_b|)/(1 - \gamma)$ and $C_b$ is the baseline reward.*

Proposition 4 is important for deriving a fast convergence rate and its proof can be found in Xu et al. [19]. Next, we need an assumption on the variance of the importance weights (Section 3).

**Assumption 5** (On variance of the importance weights). *There exists a constant $W < \infty$ such that for each policy pairs in Algorithm 1, it holds*

$$\mathrm{Var}(\omega(\tau|\boldsymbol{\theta}_1, \boldsymbol{\theta}_2)) \leq W, \qquad \forall \boldsymbol{\theta}_1, \boldsymbol{\theta}_2 \in \mathbb{R}^d, \tau \sim p(\cdot|\boldsymbol{\theta}_1)$$

Assumption 5 has also been made by Papini et al. [18], Xu et al. [19]. Now we present the convergence guarantees for our FedPG-BR algorithm:

**Theorem 6** (Convergence of FedPG-BR). *Assume uniform initial state distribution across agents, and the gradient estimator is set to be the REINFORCE or GPOMDP estimator. Under Assumptions 2, 3, and 5, if we choose $\eta_t \leq \frac{1}{2\Psi B_t^{2/3}}$, $b_t = 1$, and $B_t = B \geq 4\Phi L^{-2}$ where $\Phi \triangleq L_g + C_g^2 C_w$, $\Psi \triangleq (L(L_g + C_g^2 C_w))^{1/3}$, $L, L_g, C_g$ are defined in Proposition 4 and $C_w$ is defined in Lemma 9, $\delta \in (0, 1)$ such that $e^{\frac{\delta B_t}{2(1-2\delta)}} \leq \frac{2K}{\delta} \leq e^{\frac{B_t}{2}}$ and $\delta \leq \frac{1}{5KB_t}$, then the output $\tilde{\boldsymbol{\theta}}_a$ of Algorithm 1 satisfies*

$$\mathbb{E}[\|\nabla J(\tilde{\boldsymbol{\theta}}_a)\|^2] \leq \frac{2\Psi \left[ J(\tilde{\boldsymbol{\theta}}^*) - J(\tilde{\boldsymbol{\theta}}_0) \right]}{TB^{1/3}} + \frac{8\sigma^2}{(1-\alpha)^2 KB} + \frac{96\alpha^2\sigma^2 V}{(1-\alpha)^2 B}$$

*where $0 \leq \alpha < 0.5$ and $\tilde{\boldsymbol{\theta}}^*$ is a global maximizer of J.*

This theorem leads to many interesting insights. When $K = 1, \alpha = 0$, Theorem 6 reduces to $\mathbb{E}\|\nabla J(\tilde{\boldsymbol{\theta}}_a)\|^2 \leq 2\Psi[J(\tilde{\boldsymbol{\theta}}^*) - J(\tilde{\boldsymbol{\theta}}_0)]/TB^{1/3} + 8\sigma^2/B$. The second term here $O(1/B)$, which also shows up in SVRPG [18, 19], results from the full gradient approximation in Equation (2) in each round. In this case, our theorem implies that $\mathbb{E}\|\nabla J(\tilde{\boldsymbol{\theta}}_a)\|^2 = O(\Psi[J(\tilde{\boldsymbol{\theta}}^*) - J(\tilde{\boldsymbol{\theta}}_0)]/TB^{1/3})$ which is consistent with SCSG for $L$-smooth non-convex objective functions [35]. Moreover, using $\mathbb{E}[Traj(\epsilon)]$ to denote the expected number of trajectories required by each agent to achieve $\mathbb{E}[\|\nabla J(\tilde{\boldsymbol{\theta}}_a)\|^2] \leq \epsilon$, Theorem 6 leads to:

**Corollary 7** (Sample complexity of FedPG-BR). *Under the same assumptions as Theorem 6, let $\epsilon > 0$, we have: (i) $\mathbb{E}[Traj(\epsilon)] = O(\frac{1}{\epsilon^{5/3}K^{2/3}} + \frac{\alpha^{4/3}}{\epsilon^{5/3}})$; (ii) When $\alpha = 0$, we have $\mathbb{E}[Traj(\epsilon)] = O(\frac{1}{\epsilon^{5/3}K^{2/3}})$; (iii) When $K = 1$, we have $\mathbb{E}[Traj(\epsilon)] = O(\frac{1}{\epsilon^{5/3}})$*

We present a straightforward comparison of the sample complexity of related works in Table 1. Both REINFORCE and GPOMDP have a sample complexity of $O(1/\epsilon^2)$ since they use stochastic gradient-based optimization. Xu et al. [19] has made a refined analysis of SVRPG to improve its sample

Table 1: Sample complexities of relevant works to achieve $\mathbb{E}\|\nabla J(\boldsymbol{\theta})\|^2 \leq \epsilon$.

| SETTINGS | METHODS | COMPLEXITY |
|---|---|---|
| | REINFORCE [39] | $O(1/\epsilon^2)$ |
| | GPOMDP [40] | $O(1/\epsilon^2)$ |
| $K = 1$ | SVRPG [18] | $O(1/\epsilon^2)$ |
| | SVRPG [19] | $O(1/\epsilon^{5/3})$ |
| | **FedPG-BR** | $O(1/\epsilon^{5/3})$ |
| $K > 1, \alpha = 0$ | **FedPG-BR** | $O(\frac{1}{\epsilon^{5/3}K^{2/3}})$ |
| $K > 1, \alpha > 0$ | **FedPG-BR** | $O(\frac{1}{\epsilon^{5/3}K^{2/3}} + \frac{\alpha^{4/3}}{\epsilon^{5/3}})$ |

complexity from $O(1/\epsilon^2)$ [18] to $O(1/\epsilon^{5/3})$. Corollary 7 *(iii)* reveals that the sample complexity of FedPG-BR in the single-agent setup agrees with that of SVRPG derived by Xu et al. [19].

When $K > 1$, $\alpha = 0$, Corollary 7 *(ii)* implies that the total number of trajectories required by each agent is upper-bounded by $O(1/(\epsilon^{5/3}K^{2/3}))$. This result gives us the theoretical grounds to encourage more agents to participate in the federation, since the number of trajectories each agent needs to sample decays at a rate of $O(1/K^{2/3})$. This guaranteed improvement in sample efficiency is highly desirable in practical systems with a large number of agents.

Next, for a more realistic system where an $\alpha$-fraction ($\alpha > 0$) of the agents are Byzantine agents, Corollary 7 *(i)* assures us that the total number of trajectories required by each agent will be increased by only an additive term of $O(\alpha^{4/3}/\epsilon^{5/3})$. This term is unavoidable due to the presence of Byzantine agents in FRL systems. However, the bound implies that the impact of Byzantine agents on the overall convergence is limited, which aligns with the discussions on our Byzantine filtering strategy (Section 3.3), and will be empirically verified in our experiments. Moreover, the impact of Byzantine agents on the convergence vanishes when $\alpha \to 0$. That is, when the system is ideal ($\alpha = 0$), our Byzantine filtering step induces no effect on the convergence.

## 5 Experiments

We evaluate the empirical performances of FedPG-BR with and without Byzantine agents on different RL benchmarks, including CartPole balancing [55], LunarLander, and the 3D continuous locomotion control task of Half-Cheetah [56]. In all experiments, we measure the performance online such that in each iteration, we evaluate the current policy of the server by using it to independently interact with the test MDP for 10 trajectories and reporting the mean returns. Each experiment is independently repeated 10 times with different random seeds and policy initializations, both of which are shared among all algorithms for fair comparisons. The results are averaged over the 10 independent runs with 90% bootstrap confidence intervals. Due to space constraints, some experimental details are deferred to Appendix F.

**Performances in ideal systems with $\alpha = 0$.** We firstly evaluate the performances of FedPG-BR in ideal systems with $\alpha = 0$, i.e., no Byzantine agents. We compare FedPG-BR ($K = 1, 3, 10$) with vanilla policy gradient using GPOMDP[3] and SVRPG. The results in all three tasks are plotted in Fig. 1. The figures show that FedPG-BR ($K = 1$) and SVRPG perform comparably, both outperforming GPOMDP. This aligns with the results in Table 1 showing that FedPG-BR ($K = 1$) and SVRPG share the same sample complexity, and both are provably more sample-efficient than GPOMDP. Moreover, the performance of FedPG-BR is improved significantly with the federated of only $K = 3$ agents, and improved even further when $K = 10$. This corroborates our theoretical insights implying that the federation of more agents (i.e., larger $K$) improves the sample efficiency of FedPG-BR (Section 4), and verifies the practical performance benefit offered by the participation of more agents.

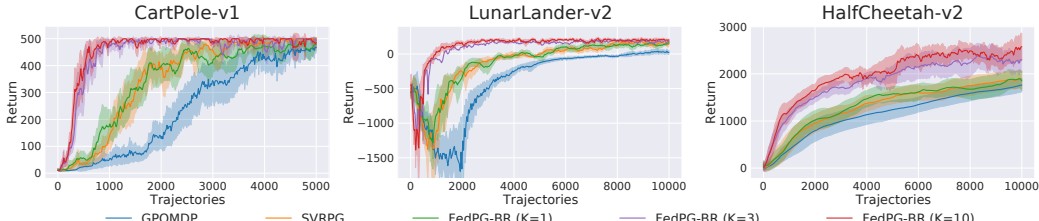

Figure 1: Performance of FedPG-BR in ideal systems with $\alpha = 0$ for the three tasks.

**Performances in practical systems with $\alpha > 0$.** Next, we investigate the impact of Byzantine agents (i.e., random failures or adversarial attacks) on the sample efficiency, which is critical for the practical deployment of FRL algorithms. In this experiment, we use $K = 10$ agents among which 3 are Byzantine agents, and we simulate different types of Byzantine failures: (a) *Random Noise (RN)*: each Byzantine agent sends a random vector to the server; (b) *Random Action (RA)*: every Byzantine agent ignores the policy from the server and takes actions randomly, which is used to

---

[3]Since GPOMDP has been repeatedly found to be comparable to or better than REINFORCE [18, 19].

simulate random system failures (e.g., hardware failures) and results in false gradient computations since the trajectories are no longer sampled according to the policy; (c) *Sign Filliping (SF)*: each Byzantine agent computes the correct gradient but sends the scaled negative gradient (multiplied by $-2.5$),which is used to simulate adversarial attacks aiming to manipulate the direction of policy update at the server.

For comparison, we have adapted both GPOMDP and SVRPG to the FRL setting (pseudocode is provided in Appendix A.3). Fig. 2 shows the results using the HalfCheetah task. We have also included the performances of GPOMDP, SVRPG and FedPG-BR in the single-agent settting ($K=1$) as dotted-line (mean value of 10 independent runs) for reference. The figures show that for both GPOMDP and SVRPG, the 3 Byzantine agents cause the performance of their federated versions to be worse than that in the single-agent setting. Particularly, RA agents (middle figure) render GPOMDP and SVRPG unlearnable, i.e., unable to converge at all. In contrast, our FedPG-BR is robust against all three types of Byzantine failures. That is, FedPG-BR ($K=10$ $B=3$) with 3 Byzantine agents still significantly outperforms the single-agent setting, and more importantly, *performs comparably to FedPG-BR ($K=10$) with 10 good agents*. This is because our Byzantine filtering strategy can effectively filter out those Byzantine agents. These results demonstrate that even in practical systems which are subject to random failures or adversarial attacks, FedPG-BR is still able to deliver superior performances. This provides an assurance on the reliability of our FedPG-BR algorithm to promote its practical deployment, and significantly improves the practicality of FRL. The results for the CartPole and LunarLander tasks, which yield the same insights as discussed here, can be found in Appendix G.

**Performance of FedPG-BR against FedPG attack.** We have discussed (Section 3.3) and shown through theoretical analysis (Section 4) that even when our Byzantine filtering strategy fails, the impact of the Byzantine agents on the performance of our algorithm is still limited. Here we verify this empirically. To this end, we design a new type of Byzantine agents who have perfect knowledge about our Byzantine filtering strategy, and call it *FedPG attacker*. The goal of FedPG attackers are to collude with each other to attack our algorithm without being filtered out. To achieve this, FedPG attackers firstly estimate $\nabla J(\boldsymbol{\theta}_0^t)$ using the mean of their gradients $\bar{\mu}_t$, and estimate $\sigma$ by calculating the maximum Euclidean distance between the gradients of any two FedPG attackers as $2\bar{\sigma}$. Next, all FedPG attackers send the vector $\bar{\mu}_t + 3\bar{\sigma}$ to the server. Recall we have discussed in Section 3.3 that if a Byzantine agent is not filtered out, its distance to the true gradient $\nabla J(\boldsymbol{\theta}_0^t)$ is at most $3\sigma$. Therefore, if $\bar{\mu}_t$ and $\bar{\sigma}$ are estimated accurately, the vectors from the FedPG attackers can exert negative impact on the convergence while still evading our Byzantine filtering.

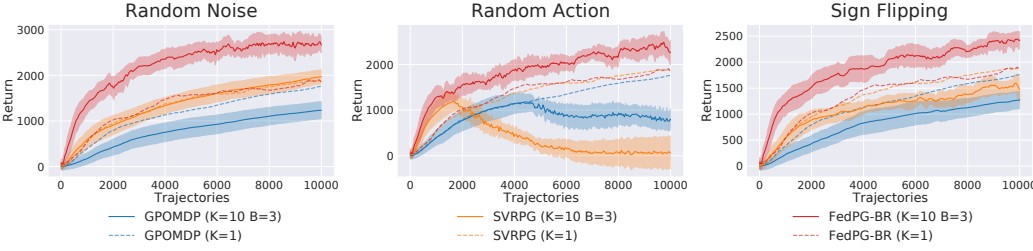

Figure 2: Performance of FedPG-BR in practical systems with $\alpha > 0$ for HalfCheetah. Each subplot corresponds to a different type of Byzantine failure exercised by the 3 Byzantine agents.

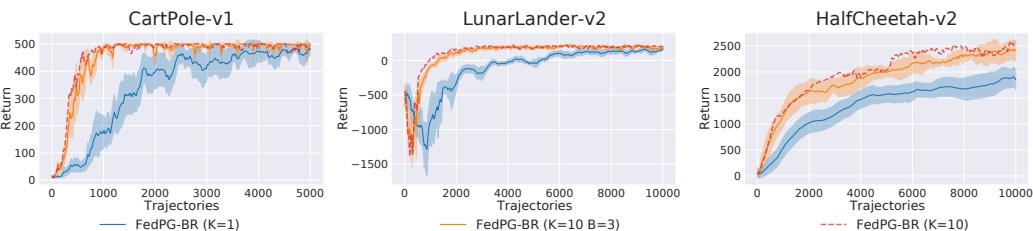

Figure 3: Performance of FedPG-BR in practical systems against FedPG attack.

We again use $K = 10$ agents, among which 3 are FedPG attackers. The results (Fig. 3) show that in all three tasks, even against such strong attackers with perfect knowledge of our Byzantine filtering strategy, FedPG-BR ($K = 10$ $B = 3$) still manages to significantly outperform FedPG-BR ($K = 1$) in the single-agent setting. Moreover, the performance of FedPG-BR ($K = 10$ $B = 3$) is only marginally worsened compared with FedPG-BR ($K = 10$) with 10 good agents. This corroborates our theoretical analysis showing that although we place no assumption on the gradients sent by the Byzantine agents, they only contribute an additive term of $O(\alpha^{4/3}/\epsilon^{5/3})$ to the sample complexity (Section 4). These results demonstrate the empirical robustness of FedPG-BR even against strong attackers, hence further highlighting its practical reliability.

## 6 Conclusion and future work

Federation is promising in boosting the sample efficiency of RL agents, without sharing their trajectories. Due to the high sampling cost of RL, the design of FRL systems appeals for theoretical guarantee on its convergence which is, however, vulnerable to failures and attacks in practical setup, as demonstrated. This paper provides the theoretical ground to study the sample efficiency of FRL with respect to the number of participating agents, while accounting for faulty agents. We verify the empirical efficacy of the proposed FRL framework in systems with and without different types of faulty agents on various RL benchmarks.

Variance control is the key to exploiting Assumption 2 on the bounded variance of PG estimators in our filter design. As a result, our framework is restricted to the variance-reduced policy gradient methods. Intuitively, it is worth studying the fault-tolerant federation of other policy optimization methods. Another limitation of this work is that agents are assumed to be homogeneous, while in many real-world scenarios, RL agents are heterogeneous. Therefore, it would be interesting to explore the possibility of heterogeneity of agents in fault-tolerant FRL in future works. Moreover, another interesting future work is to apply our Byzantine filtering strategy to other federated sequential decision-making problems such as federated bandit [57–59] and federated/collaborative Bayesian optimization [60–62], as well as other settings of collaborative multi-party ML [63–75], to equip them with theoretically guaranteed fault tolerance.

## Acknowledgments and Disclosure of Funding

This research/project is supported by A*STAR under its RIE2020 Advanced Manufacturing and Engineering (AME) Industry Alignment Fund – Pre Positioning (IAF-PP) (Award A19E4a0101) and its A*STAR Computing and Information Science Scholarship (ACIS) awarded to Flint Xiaofeng Fan. Wei Jing is supported by Alibaba Innovative Research (AIR) Program.

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
