# A   More on the background

## A.1   SVRG and SCSG

Here we provide the pseudocode for SVRG (Algorithm 2) and SCSG (Algorithm 3) seen in Lei et al. [35]. The idea of SVRG (Algorithm 2) is to reuses past *full* gradient computations (line 3) to reduce the variance of the current *stochastic* gradient estimate (line 7) before the parameter update (line 8). Note that $N = 1$ corresponds to a GD step (i.e., $v_{k-1}^{(j)} \leftarrow g_j$ in line 7). For $N > 1$, $v_{k-1}^{(j)}$ is the corrected gradient in SVRG and is an unbiased estimate of the true gradient $\nabla J(\boldsymbol{\theta})$. SVRG achieves linear convergence $O(1/T)$ using the semi-stochastic gradient.

---

**Algorithm 2** SVRG

1: **Input:** Number of stages $T$, initial iteratre $\tilde{\boldsymbol{\theta}}_0$, number of gradient steps $N$, step size $\eta$
2: **for** $t = 1$ **to** $T$ **do**
3:     $g_t \leftarrow \nabla J(\tilde{\boldsymbol{\theta}}_{t-1}) = \frac{1}{n} \sum_{i=1}^{n} \nabla J_i(\tilde{\boldsymbol{\theta}}_{t-1})$
4:     $\boldsymbol{\theta}_0^{(t)} \leftarrow \tilde{\boldsymbol{\theta}}_{t-1}$
5:     **for** $k = 1$ **to** $N$ **do**
6:        Randomly pick $i_k \in [n]$
7:        $v_{k-1}^{(t)} \leftarrow \nabla J_{i_k}(\boldsymbol{\theta}_{k-1}^{(t)}) - \nabla J_{i_k}(\boldsymbol{\theta}_0^{(t)}) + g_t$
8:        $\boldsymbol{\theta}_k^{(t)} \leftarrow \boldsymbol{\theta}_{k-1}^{(t)} - \eta_t v_{k-1}^{(t)}$
9:     $\tilde{\boldsymbol{\theta}}_t \leftarrow \boldsymbol{\theta}_{N_t}^{(t)}$
10: **Output:** $\tilde{\boldsymbol{\theta}}_T$ (Convex case) or $\tilde{\boldsymbol{\theta}}_t$ uniformly picked from $\{\tilde{\boldsymbol{\theta}}_t\}_{t=1}^{T}$ (Non-Convex case)

---

More recently, *Stochastically Controlled Stochastic Gradient* (SCSG) has been proposed [34], to further reduce the computational cost of SVRG. The key difference is that SCSG (Algorithm 3) considers a sequence of time-varying batch sizes ($B_t$ and $b_t$) and employs geometric sampling to generate the number of parameter update steps $N_t$ in each iteration (line 6), instead of fixing the batch sizes and the number of updates as done in SVRG. Particularly when finding an $\epsilon$-approximate solution (Definition 1) for optimizing smooth non-convex objectives, Lei et al. [35] proves that SCSG is never worse than SVRG in convergence rate and significantly outperforms SVRG when the required $\epsilon$ is small.

---

**Algorithm 3** SCSG for smooth non-convex objectives

1: **Input:** Number of stages $T$, initial iteratre $\tilde{\boldsymbol{\theta}}_0$, batch size $B_t$, mini-batch size $b_t$, step size $\eta_t$
2: **for** $t = 1$ **to** $T$ **do**
3:     Uniformly sample a batch $\mathcal{I}_t \subset \{1, \cdots, n\}$ with $|\mathcal{I}_t| = B_t$
4:     $g_t \leftarrow \nabla J_{\mathcal{I}_t}(\tilde{\boldsymbol{\theta}}_{t-1})$
5:     $\boldsymbol{\theta}_0^{(t)} \leftarrow \tilde{\boldsymbol{\theta}}_{t-1}$
6:     Generate $N_t \sim \text{Geom}(B_t/(B_t + b_t))$
7:     **for** $k = 1$ **to** $N_t$ **do**
8:        Randomly pick $\tilde{\mathcal{I}}_{k-1} \subset [n]$ with $|\tilde{\mathcal{I}}_{k-1}| = b_t$
9:        $v_{k-1}^{(t)} \leftarrow \nabla J_{\tilde{\mathcal{I}}_{k-1}}(\boldsymbol{\theta}_{k-1}^{(t)}) - \nabla J_{\tilde{\mathcal{I}}_{k-1}}(\boldsymbol{\theta}_0^{(t)}) + g_t$
10:       $\boldsymbol{\theta}_k^{(t)} \leftarrow \boldsymbol{\theta}_{k-1}^{(t)} + \eta_t v_{k-1}^{(t)}$
11:     $\tilde{\boldsymbol{\theta}}_t \leftarrow \boldsymbol{\theta}_{N_t}^{(t)}$
12: **Output:** $\tilde{\boldsymbol{\theta}}_T$ (P-L case) or sample $\tilde{\boldsymbol{\theta}}_T^*$ from $\{\tilde{\boldsymbol{\theta}}_t\}_{t=1}^{T}$ with $P(\tilde{\boldsymbol{\theta}}_T^* = \tilde{\boldsymbol{\theta}}_t) \propto \eta_t B_t/b_t$ (Smooth case)

---

As a member of the SVRG-like algorithms, SCSG enjoys the same convergence rate of SVRG while being computationally cheaper than SVRG for tasks with small $\epsilon$ requirements [34], which is highly desired in RL, hence the motivation of FedPG-BR to adapt SCSG.

## A.2 Gradient estimator

Use $g(\tau|\boldsymbol{\theta})$ to denote the *unbiased* estimator to the true gradient $J(\boldsymbol{\theta})$. The common gradient estimators are the REINFORCE and the GPOMDP estimators, which are considered as baseline estimators in [18] and [19]. The REINFORCE [39]:

$$g(\tau|\boldsymbol{\theta}) = \left[\sum_{h=0}^{H-1} \nabla_{\boldsymbol{\theta}} \log \pi_{\boldsymbol{\theta}}(a_h \mid s_h)\right]\left[\sum_{h=0}^{H-1} \gamma^h \mathcal{R}(s_h, a_h) - C_b\right]$$

And the GPOMDP [40]

$$g(\tau|\boldsymbol{\theta}) = \sum_{h=0}^{H-1}\left[\sum_{t=0}^{h} \nabla_{\boldsymbol{\theta}} \log \pi_{\boldsymbol{\theta}}(a_t \mid s_t)\right](\gamma^h r(s_h, a_h) - C_{b_h})$$

where $C_b$ and $C_{b_h}$ are the corresponding baselines. Under Assumption 3, whether we use the REINFORCE or the GPOMDP estimator, Proposition 4 holds [18, 19].

---

**Algorithm 4** GPOMDP (for federation of K agents)

---

**Input:** number of iterations $T$, batch size $B$, step size $\eta$, initial parameter $\tilde{\boldsymbol{\theta}}_0 \in \mathbb{R}^d$
**for** $t = 1$ **to** $T$ **do**
   $\boldsymbol{\theta}^t \leftarrow \tilde{\boldsymbol{\theta}}_{t-1}$                  ; broadcast to agents
   **for** $k = 1$ **to** $K$ **do**
      Sample $B$ trajectories $\{\tau_{t,i}^{(k)}\}$ from $p(\cdot|\boldsymbol{\theta}^t)$
      $\mu_t^{(k)} = \frac{1}{B}\sum_{i=1}^{B} g(\tau_{t,i}^{(k)}|\boldsymbol{\theta}^t)$        ; push $\mu_t^{(k)}$ to server
   $\mu_t = \frac{1}{K}\sum_{k=1}^{K} \mu_t^{(k)}$
   $\tilde{\boldsymbol{\theta}}_t \leftarrow \boldsymbol{\theta}^t + \eta\mu_t$
**Output** $\boldsymbol{\theta}_{out}$: uniformly randomly picked from $\{\tilde{\boldsymbol{\theta}}_t\}_{t=1}^{T}$

---

## A.3 Federated GPOMDP and SVRPG

Closely following the problem setting of FedPG-BR, we adapt both GPOMDP and SVRPG to the FRL setting. The pseudocode is shown in Algorithm 4 and Algorithm 5.

---

**Algorithm 5** SVRPG (for federation of K agents)

---

**Input:** number of epochs $T$, epoch size $N$, batch size $B$, mini-batch size $b$, step size $\eta$, initial parameter $\tilde{\boldsymbol{\theta}}_0 \in \mathbb{R}^d$
**for** $t = 1$ **to** $T$ **do**
   $\boldsymbol{\theta}_0^t \leftarrow \tilde{\boldsymbol{\theta}}_{t-1}$                  ; broadcast to agents
   **for** $k = 1$ **to** $K$ **do**
      Sample $B$ trajectories $\{\tau_{t,i}^{(k)}\}$ from $p(\cdot|\boldsymbol{\theta}_0^t)$
      $\mu_t^{(k)} = \frac{1}{B}\sum_{i=1}^{B} g(\tau_{t,i}^{(k)}|\boldsymbol{\theta}_0^t)$      ; push $\mu_t^{(k)}$ to server
   $\mu_t = \frac{1}{K}\sum_{k=1}^{K} \mu_t^{(k)}$
   **for** $n = 0$ **to** $N - 1$ **do**
      Sample $b$ trajectories $\{\tau_{n,j}^t\}$ from $p(\cdot|\boldsymbol{\theta}_n^t)$
      $v_n^t = \frac{1}{b}\sum_{j=1}^{b}[g(\tau_{n,j}^t|\boldsymbol{\theta}_n^t) - \omega(\tau_{n,j}^t|\boldsymbol{\theta}_n^t, \boldsymbol{\theta}_0^t)g(\tau_{n,j}^t|\boldsymbol{\theta}_0^t)] + \mu_t$
      $\boldsymbol{\theta}_{n+1}^t = \boldsymbol{\theta}_n^t + \eta v_n^t$
   $\tilde{\boldsymbol{\theta}}_t \leftarrow \boldsymbol{\theta}_N^t$
**Output** $\boldsymbol{\theta}_{out}$: uniformly randomly picked from $\{\tilde{\boldsymbol{\theta}}_t\}_{t=1}^{T}$

---

# B Proof of Theorem 6

In our proof, we follow the suggestion from Lei et al. [35] to set $b_t = 1$ to derive better theoretical results. Refer to Section F in this appendix for the value of $b_t$ used in our experiments.

*Proof.* From the L-smoothness of the objective function $J(\boldsymbol{\theta})$, we have

$$\mathbb{E}_{\tau_n^t}[J(\boldsymbol{\theta}_{n+1}^t)] \geq \mathbb{E}_{\tau_n^t}\left[J(\boldsymbol{\theta}_n^t) + \langle \nabla J(\boldsymbol{\theta}_n^t), \boldsymbol{\theta}_{n+1}^t - \boldsymbol{\theta}_n^t \rangle - \frac{L}{2}\|\boldsymbol{\theta}_{n+1}^t - \boldsymbol{\theta}_n^t\|^2\right]$$

$$= J(\boldsymbol{\theta}_n^t) + \eta_t\langle \mathbb{E}_{\tau_n^t}[v_n^t], \nabla J(\boldsymbol{\theta}_n^t)\rangle - \frac{L\eta_t^2}{2}\mathbb{E}_{\tau_n^t}[\|v_n^t\|^2]$$

$$\geq J(\boldsymbol{\theta}_n^t) + \eta_t\langle \nabla J(\boldsymbol{\theta}_n^t) + e_t, \nabla J(\boldsymbol{\theta}_n^t)\rangle$$
$$- \frac{L\eta_t^2}{2}[(2L_g + 2C_g^2 C_w)\|\boldsymbol{\theta}_n^t - \boldsymbol{\theta}_0^t\|^2 + 2\|\nabla J(\boldsymbol{\theta}_n^t)\|^2 + 2\|e_t\|^2] \quad (4)$$

$$= J(\boldsymbol{\theta}_n^t) + \eta_t(1 - L\eta_t)\|\nabla J(\boldsymbol{\theta}_n^t)\|^2 + \eta_t\langle e_t, \nabla J(\boldsymbol{\theta}_n^t)\rangle$$
$$- L\eta_t^2(L_g + C_g^2 C_w)\|\boldsymbol{\theta}_n^t - \boldsymbol{\theta}_0^t\|^2 - L\eta_t^2\|e_t\|^2$$

where (4) follows from Lemma 11. Use $\mathbb{E}_t$ to denote the expectation with respect to all trajectories $\{\tau_1^t, \tau_2^t, ...\}$, given $N_t$. Since $\{\tau_1^t, \tau_2^t, ...\}$ are independent of $N_t$, $\mathbb{E}_t$ is equivalently the expectation with respect to $\{\tau_1^t, \tau_2^t, ...\}$. The above inequality gives

$$\mathbb{E}_t[J(\boldsymbol{\theta}_{n+1}^t)] \geq \mathbb{E}_t[J(\boldsymbol{\theta}_n^t)] + \eta_t(1 - L\eta_t)\mathbb{E}_t\|\nabla J(\boldsymbol{\theta}_n^t)\|^2 + \eta_t\mathbb{E}_t\langle e_t, \nabla J(\boldsymbol{\theta}_n^t)\rangle$$
$$- L\eta_t^2(L_g + C_g^2 C_w)\mathbb{E}_t\|\boldsymbol{\theta}_n^t - \boldsymbol{\theta}_0^t\|^2 - L\eta_t^2\|e_t\|^2$$

Taking $n = N_t$ and using $\mathbb{E}_{N_t}$ to denote the expectation w.r.t. $N_t$, we have from the above

$$\mathbb{E}_{N_t}\mathbb{E}_t[J(\boldsymbol{\theta}_{N_t+1}^t)] \geq \mathbb{E}_{N_t}\mathbb{E}_t[J(\boldsymbol{\theta}_{N_t}^t)] + \eta_t(1 - L\eta_t)\mathbb{E}_{N_t}\mathbb{E}_t\|\nabla J(\boldsymbol{\theta}_{N_t}^t)\|^2 + \eta_t\mathbb{E}_{N_t}\mathbb{E}_t\langle e_t, \nabla J(\boldsymbol{\theta}_{N_t}^t)\rangle$$
$$- L\eta_t^2(L_g + C_g^2 C_w)\mathbb{E}_{N_t}\mathbb{E}_t\|\boldsymbol{\theta}_{N_t}^t - \boldsymbol{\theta}_0^t\|^2 - L\eta_t^2\|e_t\|^2$$

Rearrange,

$$\eta_t(1 - L\eta_t)\mathbb{E}_{N_t}\mathbb{E}_t\|\nabla J(\boldsymbol{\theta}_{N_t}^t)\|^2 \leq \mathbb{E}_{N_t}\mathbb{E}_t[J(\boldsymbol{\theta}_{N_t+1}^t)] + L\eta_t^2(L_g + C_g^2 C_w)\mathbb{E}_{N_t}\mathbb{E}_t\|\boldsymbol{\theta}_{N_t}^t - \boldsymbol{\theta}_0^t\|^2$$
$$- \eta_t\mathbb{E}_{N_t}\mathbb{E}_t\langle e_t, \nabla J(\boldsymbol{\theta}_{N_t}^t)\rangle + L\eta_t^2\|e_t\|^2 - \mathbb{E}_{N_t}\mathbb{E}_t[J(\boldsymbol{\theta}_{N_t}^t)]$$

$$= \frac{1}{B_t}(\mathbb{E}_t\mathbb{E}_{N_t}[J(\boldsymbol{\theta}_{N_t}^t)] - J(\boldsymbol{\theta}_0^t)) - \eta_t\mathbb{E}_{N_t}\mathbb{E}_t\langle e_t, \nabla J(\boldsymbol{\theta}_{N_t}^t)\rangle$$
$$+ L\eta_t^2(Lg + C_g^2 C_w)\mathbb{E}_{N_t}\mathbb{E}_t\left\|\boldsymbol{\theta}_{N_t}^t - \boldsymbol{\theta}_0^t\right\|^2 + L\eta_t^2\left\|e_t\right\|^2 \quad (5)$$

where (5) follows from Lemma 16 with Fubini's theorem. Note that $\tilde{\boldsymbol{\theta}}_t = \boldsymbol{\theta}_{N_t}^t$ and $\tilde{\boldsymbol{\theta}}_{t-1} = \boldsymbol{\theta}_0^t$. If we take expectation over all the randomness and denote it by $\mathbb{E}$, we get

$$\eta_t(1 - L\eta_t)\mathbb{E}\|\nabla J(\tilde{\boldsymbol{\theta}}_t)\|^2 = \frac{1}{B_t}\mathbb{E}\left[J(\tilde{\boldsymbol{\theta}}_t) - J(\tilde{\boldsymbol{\theta}}_{t-1})\right] - \eta_t\mathbb{E}\left\langle e_t, \nabla J(\tilde{\boldsymbol{\theta}}_t)\right\rangle$$
$$+ L\eta_t^2(L_g + C_g^2 C_w)\mathbb{E}\|\tilde{\boldsymbol{\theta}}_t - \tilde{\boldsymbol{\theta}}_{t-1}\|^2 + L\eta_t^2\mathbb{E}\|e_t\|^2$$

$$= \frac{1}{B_t}\mathbb{E}\left[J(\tilde{\boldsymbol{\theta}}_t) - J(\tilde{\boldsymbol{\theta}}_{t-1})\right] - \frac{1}{B_t}\mathbb{E}\left\langle e_t, \tilde{\boldsymbol{\theta}}_t - \tilde{\boldsymbol{\theta}}_{t-1}\right\rangle$$
$$+ L\eta_t^2(L_g + C_g^2 C_w)\mathbb{E}\|\tilde{\boldsymbol{\theta}}_t - \tilde{\boldsymbol{\theta}}_{t-1}\|^2 + \eta_t(1 + L\eta_t)\mathbb{E}\|e_t\|^2 \quad (6)$$

$$\leq \frac{1}{B_t}\mathbb{E}\left[J(\tilde{\boldsymbol{\theta}}_t) - J(\tilde{\boldsymbol{\theta}}_{t-1})\right]$$
$$+ \frac{1}{2\eta_t B_t}[-\frac{1}{B_t} + \eta_t^2(2L_g + 2C_g^2 C_w)]\mathbb{E}\|\tilde{\boldsymbol{\theta}}_t - \tilde{\boldsymbol{\theta}}_{t-1}\|^2$$
$$+ \frac{1}{B_t}\mathbb{E}\left\langle \nabla J(\tilde{\boldsymbol{\theta}}_t), \tilde{\boldsymbol{\theta}}_t - \tilde{\boldsymbol{\theta}}_{t-1}\right\rangle + \frac{\eta_t}{B_t}\mathbb{E}\|\nabla J(\tilde{\boldsymbol{\theta}}_t)\|^2 + \frac{\eta_t}{B_t}\mathbb{E}\|e_t\|^2$$
$$+ L\eta_t^2(L_g + C_g^2 C_w)\mathbb{E}\|\tilde{\boldsymbol{\theta}}_t - \tilde{\boldsymbol{\theta}}_{t-1}\|^2 + \eta_t(1 + L\eta_t)\mathbb{E}\|e_t\|^2 \quad (7)$$

where (6) follows from Lemma 12 and (7) follows from Lemma 13. Rearrange,

$$\eta_t(1 - L\eta_t - \frac{1}{B_t})\mathbb{E}\|\nabla J(\tilde{\boldsymbol{\theta}}_t)\|^2 + \frac{1 - 2\eta_t^2(L_g + C_g^2 C_w)B_t - 2L\eta_t^3(L_g + C_g^2 C_w)B_t^2}{2\eta_t B_t^2}\mathbb{E}\|\tilde{\boldsymbol{\theta}}_t - \tilde{\boldsymbol{\theta}}_{t-1}\|^2$$

$$\leq \frac{1}{B_t}\mathbb{E}\left[J(\tilde{\boldsymbol{\theta}}_t) - J(\tilde{\boldsymbol{\theta}}_{t-1})\right] + \frac{1}{B_t}\mathbb{E}\left\langle \nabla J(\tilde{\boldsymbol{\theta}}_t), \tilde{\boldsymbol{\theta}}_t - \tilde{\boldsymbol{\theta}}_{t-1}\right\rangle + \eta_t(1 + L\eta_t + \frac{1}{B_t})\mathbb{E}\|e_t\|^2 \quad (8)$$

Now we can apply Lemma 17 on $\mathbb{E}\left\langle \nabla J(\tilde{\boldsymbol{\theta}}_t), \tilde{\boldsymbol{\theta}}_t - \tilde{\boldsymbol{\theta}}_{t-1} \right\rangle$ using $a = \tilde{\boldsymbol{\theta}}_t - \tilde{\boldsymbol{\theta}}_{t-1}$, $b = \nabla J(\tilde{\boldsymbol{\theta}}_t)$, and $\beta = \frac{1 - 2\eta_t^2(L_g + C_g^2 C_w)B_t - 2L\eta_t^3(L_g + C_g^2 C_w)B_t^2}{\eta_t B_t}$ to get

$$\frac{1}{B_t}\mathbb{E}\left\langle \nabla J(\tilde{\boldsymbol{\theta}}_t), \tilde{\boldsymbol{\theta}}_t - \tilde{\boldsymbol{\theta}}_{t-1} \right\rangle \leq \frac{1 - 2\eta_t^2(L_g + C_g^2 C_w)B_t - 2L\eta_t^3(L_g + C_g^2 C_w)B_t^2}{2\eta_t B_t^2}\mathbb{E}\|\tilde{\boldsymbol{\theta}}_t - \tilde{\boldsymbol{\theta}}_{t-1}\|^2$$
$$+ \frac{\eta_t}{2[1 - 2\eta_t^2(L_g + C_g^2 C_w)B_t - 2L\eta_t^3(L_g + C_g^2 C_w)B_t^2]}\mathbb{E}\|\nabla J(\tilde{\boldsymbol{\theta}}_t)\|^2$$
$$(9)$$

Combining (8) and (9) and rearrange, we have

$$\eta_t(1 - L\eta_t - \frac{1}{B_t} - \frac{1}{2[1 - 2\eta_t^2(L_g + C_g^2 C_w)B_t - 2L\eta_t^3(L_g + C_g^2 C_w)B_t^2]})\mathbb{E}\|\nabla J(\tilde{\boldsymbol{\theta}}_t)\|^2$$
$$\leq \frac{1}{B_t}\mathbb{E}\left[J(\tilde{\boldsymbol{\theta}}_t) - J(\tilde{\boldsymbol{\theta}}_{t-1})\right] + \eta_t(1 + L\eta_t + \frac{1}{B_t})\mathbb{E}\|e_t\|^2$$
$$\leq \frac{1}{B_t}\mathbb{E}\left[J(\tilde{\boldsymbol{\theta}}_t) - J(\tilde{\boldsymbol{\theta}}_{t-1})\right] + \eta_t(1 + L\eta_t + \frac{1}{B_t})\left[\frac{4\sigma^2}{(1-\alpha)^2 KB_t} + \frac{48\alpha^2\sigma^2 V}{(1-\alpha)^2 B_t}\right] \quad (10)$$

where (10) follows from Lemma 15. We want to choose $\eta_t$ such that $1 - 2\eta_t^2(L_g + C_g^2 C_w)B_t - 2L\eta_t^3(L_g + C_g^2 C_w)B_t^2 > 0$. Denoting $\Phi = L_g + C_g^2 C_w$, we have

$$1 - 2\eta_t^2 \Phi B_t - 2L\eta_t^3 \Phi B_t^2 \geq 0$$
$$2\eta_t^2 \Phi B_t + 2L\eta_t^3 \Phi B_t^2 \leq 1$$

We can then choose $\eta_t$ to have $2\eta_t^2 \Phi B_t \leq \frac{1}{2}$ and $2L\eta_t^3 \Phi B_t^2 \leq \frac{1}{2}$, which implies:

$$\text{(i)} \quad \eta_t \leq \frac{1}{(4\Phi B_t)^{1/2}}$$

$$\text{(ii)} \quad \eta_t \leq \frac{1}{(4L\Phi B_t^2)^{1/3}}$$

Therefore, we can choose $\eta_t \leq \frac{1}{2(L\Phi B_t^2)^{1/3}} < \frac{1}{(4L\Phi B_t^2)^{1/3}}$, and set $B_t \geq 4\Phi L^{-2}$ to ensure both condition (i) and (ii) are satisfied, together with $\frac{1}{(4L\Phi B_t^2)^{1/3}} \leq \frac{1}{(4\Phi B_t)^{1/2}}$, and $L\eta_t + \frac{1}{B_t} \leq 1$. Thus, by choosing $\eta_t \leq \frac{1}{2\Psi B_t^{2/3}}$, where $\Psi = (L\Phi)^{1/3} = (L(L_g + C_g^2 C_w))^{1/3}$, we can obtain the following from (10):

$$\eta_t \mathbb{E}\|\nabla J(\tilde{\boldsymbol{\theta}}_t)\|^2 \leq \frac{1}{B_t}\mathbb{E}\left[J(\tilde{\boldsymbol{\theta}}_t) - J(\tilde{\boldsymbol{\theta}}_{t-1})\right] + 2\eta_t\left[\frac{4\sigma^2}{(1-\alpha)^2 KB_t} + \frac{48\alpha^2\sigma^2 V}{(1-\alpha)^2 B_t}\right]$$

Replacing $\eta_t = \frac{1}{2\Psi B_t^{2/3}}$ and rearranging, we have

$$\mathbb{E}\|\nabla J(\tilde{\boldsymbol{\theta}}_t)\|^2 \leq \frac{1}{B_t \eta_t}\mathbb{E}\left[J(\tilde{\boldsymbol{\theta}}_t) - J(\tilde{\boldsymbol{\theta}}_{t-1})\right] + 2\left[\frac{4\sigma^2}{(1-\alpha)^2 KB_t} + \frac{48\alpha^2\sigma^2 V}{(1-\alpha)^2 B_t}\right]$$
$$\leq \frac{2\Psi\mathbb{E}\left[J(\tilde{\boldsymbol{\theta}}_t) - J(\tilde{\boldsymbol{\theta}}_{t-1})\right]}{B_t^{1/3}} + \frac{8\sigma^2}{(1-\alpha)^2 KB_t} + \frac{96\alpha^2\sigma^2 V}{(1-\alpha)^2 B_t}$$

Replacing $B_t$ with constant batch size $B$ and telescoping over $t = 1, 2, ..., T$, we have for $\tilde{\boldsymbol{\theta}}_a$ from our algorithm:

$$\mathbb{E}\|\nabla J(\tilde{\boldsymbol{\theta}}_a)\|^2 \leq \frac{2\Psi\mathbb{E}\left[J(\tilde{\boldsymbol{\theta}}_T) - J(\tilde{\boldsymbol{\theta}}_0)\right]}{TB^{1/3}} + \frac{8\sigma^2}{(1-\alpha)^2 KB} + \frac{96\alpha^2\sigma^2 V}{(1-\alpha)^2 B}$$
$$\leq \frac{2\Psi\left[J(\tilde{\boldsymbol{\theta}}^*) - J(\tilde{\boldsymbol{\theta}}_0)\right]}{TB^{1/3}} + \frac{8\sigma^2}{(1-\alpha)^2 KB} + \frac{96\alpha^2\sigma^2 V}{(1-\alpha)^2 B}$$

which completes the proof. $\qquad\square$

# C  Proof of Corollary 7

*Proof.* Recall $\Psi = (L(L_g + C_g^2 C_w))^{1/3}$. From Theorem 6, we have

$$\mathbb{E}\|\nabla J(\tilde{\boldsymbol{\theta}}_a)\|^2 \leq \underbrace{\frac{2\Psi\mathbb{E}\left[J(\tilde{\boldsymbol{\theta}}^*) - J(\tilde{\boldsymbol{\theta}}_0)\right]}{TB^{1/3}}}_{T=O(\frac{1}{\epsilon B^{1/3}})} + \underbrace{\frac{8\sigma^2}{(1-\alpha)^2 KB}}_{B_K=O(\frac{1}{\epsilon K})} + \underbrace{\frac{96\alpha^2\sigma^2 V}{(1-\alpha)^2 B}}_{B_\alpha=O(\frac{\alpha^2}{\epsilon})}$$

To guarantee that the output of Algorithm 1 is $\epsilon$-approximate, i.e., $\mathbb{E}\|\nabla J(\tilde{\boldsymbol{\theta}}_a)\|^2 \leq \epsilon$, we need the number of rounds $T$ and the batch size $B$ to meet the following:

$$\text{(i)}T = O(\frac{1}{\epsilon B^{1/3}}), \text{(ii)}B_K = O(\frac{1}{\epsilon K}), \text{ and (iii)}B_\alpha = O(\frac{\alpha^2}{\epsilon})$$

By union bound and using $\mathbb{E}[Traj(\epsilon)]$ to denote the total number of trajectories required by each agent to sample, the above implies that

$$\mathbb{E}[Traj(\epsilon)] \leq TB_K + TB_\alpha$$
$$\leq O(\frac{1}{\epsilon^{5/3}K^{2/3}} + \frac{\alpha^{4/3}}{\epsilon^{5/3}})$$

in order to obtain an $\epsilon$-approximate policy, which completes the proof for Corollary 7 *(i)*. Note that the total number of trajectories generated across the whole FRL system, denoted by $\mathbb{E}[Traj_{total}(\epsilon)]$ is thus bounded by:

$$\mathbb{E}[Traj_{total}(\epsilon)] \leq O(\frac{K^{1/3}}{\epsilon^{5/3}} + \frac{K\alpha^{4/3}}{\epsilon^{5/3}})$$

Now for an ideal system where $\alpha = 0$:

$$\mathbb{E}[Traj(\epsilon)] \leq O(\frac{1}{\epsilon^{5/3}K^{2/3}})$$
$$\mathbb{E}[Traj_{total}(\epsilon)] \leq O(\frac{K^{1/3}}{\epsilon^{5/3}})$$

which completes the proof for Corollary 7 *(ii)*. Moreover, when $K = 1$, the number of trajectories required by the agent using FedPG-BR is

$$\mathbb{E}[Traj(\epsilon)] \leq O(\frac{1}{\epsilon^{5/3}})$$

which is Corollary 7 *(iii)* and is coherent with the recent analysis of SVRPG [19]. $\square$

# D  More on the Byzantine Filtering Step

In this section, we continue our discussion on our *Byzantine Filtering Step* in Section 3.3. We include the pseudocode for the subroutine **FedPG-Aggregate** below for ease of reference:

As discussed in Section 3.3, R2 (line 8 in Algorithm 1.1) ensures that $\mathcal{G}_t$ always include all good agents and for any Byzantine agents being included, their impact on the convergence of Algorithm 1 is limited since their maximum distance to $\nabla J(\boldsymbol{\theta}_0^t)$ is bounded by $3\sigma$. Here we give proofs for the claims.

**Claim D.1.** *Under Assumption 2 and $\forall\alpha < 0.5$, the filtering rule R2 in Algorithm 1.1 ensures that, in any round $t$, all gradient estimates sent from non-Byzantine agents are included in $\mathcal{G}_t$, i.e., $|\mathcal{G}_t| \geq (1-\alpha)K$.*

*Proof.* First, from Assumption 2:

$$\|\mu_t^{(k)} - \nabla J(\boldsymbol{\theta}_0^t)\| \leq \sigma, \forall k \in \mathcal{G}$$

---

**Algorithm 1.1 FedPG-Aggregate**

---

1: **Input:** Gradient estimates from $K$ agents in round $t$: $\{\mu_t^{(k)}\}_{k=1}^k$, Variance Bound $\sigma$, filtering threshold $\mathfrak{T}_\mu \triangleq 2\sigma\sqrt{\frac{V}{B_t}}$, where $V \triangleq 2\log(\frac{2K}{\delta})$ and $\delta \in (0,1)$

2: $S_1 \triangleq \{\mu_t^{(k)}\}$ where $k \in [K]$ s.t. $\left|\left\{k' \in [K] : \left\|\mu_t^{(k')} - \mu_t^{(k)}\right\| \leq \mathfrak{T}_\mu\right\}\right| > \frac{K}{2}$

3: $\mu_t^{\mathrm{mom}} \leftarrow \operatorname{argmin}_{\mu_t^{(\tilde{k})}} \|\mu_t^{(\tilde{k})} - \mathrm{mean}(S_1)\|$ where $\tilde{k} \in S_1$

4: *R1:* $\mathcal{G}_t \triangleq \left\{k \in [K] : \left\|\mu_t^{(k)} - \mu_t^{\mathrm{mom}}\right\| \leq \mathfrak{T}_\mu\right\}$

5: **if** $|\mathcal{G}_t| < (1-\alpha)K$ **then**

6: $\quad S_2 \triangleq \{\mu_t^{(k)}\}$ where $k \in [K]$ s.t. $\left|\left\{k' \in [K] : \left\|\mu_t^{(k')} - \mu_t^{(k)}\right\| \leq 2\sigma\right\}\right| > \frac{K}{2}$

7: $\quad \mu_t^{\mathrm{mom}} \leftarrow \operatorname{argmin}_{\mu_t^{(\tilde{k})}} \|\mu_t^{(\tilde{k})} - \mathrm{mean}(S_2)\|$ where $\tilde{k} \in S_2$

8: $\quad$ *R2:* $\mathcal{G}_t \triangleq \left\{k \in [K] : \left\|\mu_t^{(k)} - \mu_t^{\mathrm{mom}}\right\| \leq 2\sigma\right\}$

9: **Return:** $\mu_t \triangleq \frac{1}{|\mathcal{G}_t|} \sum_{k \in \mathcal{G}_t} \mu_t^{(k)}$

---

it implies that $\|\mu_t^{(k_1)} - \mu_t^{(k_2)}\| \leq 2\sigma, \forall k_1, k_2 \in \mathcal{G}$. So, for any value of the *vector median* [46] in S2 $= \{\mu_t^{(k)}\}$ (defined in line 6):

$$\|\mu_t^{(k)} - \nabla J(\boldsymbol{\theta}_0^t)\| \leq 3\sigma, \forall \mu_t^{(k)} \in \mathrm{S2}$$

An intuitive illustration is provided in Fig. 4. Next, consider the worst case where all values sent by the $K$ agents are included in S2: for all $(1-\alpha)K$ good agents, they send the same value $\mu_t^{(k)}$, s.t., $\|\nabla J(\boldsymbol{\theta}_0^t) - \mu_t^{(k)}\| = \sigma, \forall k \in \mathcal{G}$; and for all $\alpha K$ Byzantine agents, they send the same value $\mu_t^{(k')}$ s.t., $\|\nabla J(\boldsymbol{\theta}_0^t) - \mu_t^{(k')}\| = 3\sigma, \forall k' \in \mathrm{S2} \setminus \mathcal{G}$. Then the mean of values in S2 satisfies:

$$\begin{aligned}
\|\mu_t^{\mathrm{mean}} - \nabla J(\boldsymbol{\theta}_0^t)\| &= \frac{(1-\alpha)K \cdot \sigma + \alpha K \cdot 3\sigma}{K} \\
&= (1-\alpha)\sigma + 3\alpha\sigma \\
&= \sigma + 2\alpha\sigma \\
&< 2\sigma
\end{aligned}$$

where the last inequality holds for $\alpha < 0.5$ which is our assumption. Then the value $\mu_t^{\mathrm{mom}}$ of Algorithm 1 will be set to any $\mu_t^{(k)}$ from S2, of which is the closet to $\mu_t^{\mathrm{mean}}$.

The selection of $\mu_t^{\mathrm{mom}}$ implies $\|\mu_t^{\mathrm{mom}} - \nabla J(\boldsymbol{\theta}_0^t)\| \leq \sigma$. Therefore, by constructing a region of $\mathcal{G}_t$ that is centred at $\mu_t^{\mathrm{mom}}$ and $2\sigma$ in radius (line 8), $\mathcal{G}_t$ can cover all estimates from non-Byzantine agents and hence ensure $|\mathcal{G}_t| \geq (1-\alpha)K$. $\qquad \square$

**Claim D.2.** *Under Assumption 2 and $\alpha < 0.5$, the filtering rule R2 in Algorithm 1.1 ensures that, in any round $t$, $\|\mu_t^{(k)} - \nabla J(\boldsymbol{\theta}_0^t)\| \leq 3\sigma, \forall k \in \mathcal{G}_t$.*

*Proof.* This lemma is a straightforward result following the proof of Claim D.1. $\qquad \square$

**Remark.** *Claim D.2 implies that, in any round $t$, if an estimate sent from Byzantine agent is included in $\mathcal{G}_t$, then its impact on the convergence of Algorithm 1 is limited since its distance to $\nabla J(\boldsymbol{\theta}_0^t)$ is bounded by $3\sigma$. Fig. 4 provides an intuitive illustration for this claim.*

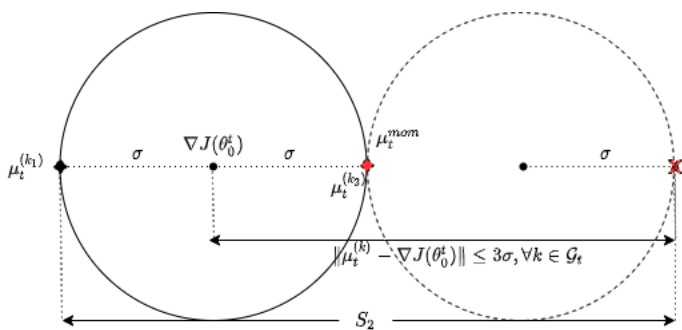

Figure 4: Graphical illustration of the Byzantine filtering strategy where $\mu_t^{(k_1)}, \mu_t^{(k_2)}$ are two good gradients while the red cross represents one Byzantine gradient which falls within $S_2$. $\mu_t^{\text{mom}}$ will be chosen at the red diamond.

As discussed above, R2 ensures that all good agents are included in $\mathcal{G}_t$, i.e., a region in which all good agents are *concentrated*. R1 (lines 2-4) is designed in a similar way and aims to improve the practical performance of FedPG-BR by exploiting Lemma 14: all good agents are *highly likely* to be *concentrated* in a much *smaller region*.

**Claim D.3.** *Define $V \triangleq 2\log(2K/\delta)$ and $\delta \in (0,1)$, the filtering R1 in Algorithm 1 ensure*

$$\|\mu_t^{(k)} - \nabla J(\boldsymbol{\theta}_0^t)\| \leq \sigma\sqrt{\frac{V}{B_t}}, \forall k \in \mathcal{G}$$

*with probability of at least $1 - \delta$.*

*Proof.* From Assumption 2, $\|\mu_t^{(k)} - \nabla J(\boldsymbol{\theta}_0^t)\| \leq \sigma, \forall k \in \mathcal{G}$. We have

$$\|\mu_t^{(k)} - \nabla J(\boldsymbol{\theta}_0^t)\| = \left\|\frac{1}{B_t}\sum_{i=1}^{B_t} g(\tau_{t,i}^{(k)}|\boldsymbol{\theta}_0^t) - \nabla J(\boldsymbol{\theta}_0^t)\right\|$$

$$= \frac{1}{B_t}\sqrt{\left\|\sum_{i=1}^{B_t} g(\tau_{t,i}^{(k)}|\boldsymbol{\theta}_0^t) - \nabla J(\boldsymbol{\theta}_0^t)\right\|^2} \tag{11}$$

Consider $X_i \triangleq g(\tau_{t,i}^{(k)}) - \nabla J(\boldsymbol{\theta}_0^t)$ and apply Lemma 14 on (11), we have

$$\Pr\left[\left\|\sum_{i=1}^{B_t} X_i\right\|^2 \leq 2\log(\frac{2}{\delta})\sigma^2 B_t\right] \geq 1 - \delta$$

$$\Pr\left[\frac{1}{B_t}\sqrt{\left\|\sum_{i=1}^{B_t} X_i\right\|^2} \leq \frac{1}{B_t}\sqrt{2\log(\frac{2}{\delta})\sigma^2 B_t}\right] \geq 1 - \delta$$

With $V \triangleq 2\log(2K/\delta)$ and $\delta \in (0,1)$, the above inequality yields the Claim. $\square$

Therefore, the first filtering R1 (lines 2-4) of FedPG-BR constructs a region of $\mathcal{G}_t$ centred at $\mu_t^{\text{mom}}$ with radius of $2\sigma\sqrt{\frac{V}{B_t}}$, which ensures in any round $t$ that, *with probability $\geq 1 - \delta$*, (a) all good agents are included in $\mathcal{G}_t$, and (b) if gradients from Byzantine agents are included in $\mathcal{G}_t$, their impact is limited since their maximum distance to $\nabla J(\boldsymbol{\theta}_0^t)$ is bounded by $3\sigma\sqrt{\frac{V}{B_t}}$ (The proof is similar to that of Claim D.2). Compared to R2, R1 can construct a *smaller* region that the server believes contains all good agents. If any Byzantine agent is included, their impact is also *smaller*, with probability of at least $1 - \delta$. Therefore, R1 is applied first such that if R1 fails (line 5) which happens with probability $< \delta$, R2 is then employed as a backup to ensure that $\mathcal{G}_t$ always includes all good agents.

# E Useful technical lemmas

**Lemma 8** (Unbiaseness of importance sampling).
$$\mathbb{E}_{\tau \sim p(\cdot|\boldsymbol{\theta}_n)}[\omega(\tau|\boldsymbol{\theta}_n, \boldsymbol{\theta}_0)g(\tau|\boldsymbol{\theta}_0)] = \mathbb{E}_{\tau \sim p(\cdot|\boldsymbol{\theta}_0)}[g(\tau|\boldsymbol{\theta}_0)]$$
$$= \nabla J(\boldsymbol{\theta}_0)$$

*Proof.* Drop $t$ from notation and use $\tau_n$ to denote trajectories sampled from $\boldsymbol{\theta}_n$ at step $n$. From the definition of gradient estimation, we have

$$g(\tau_n|\boldsymbol{\theta}_0) = \mathbb{E}_{\tau \sim p(\cdot|\boldsymbol{\theta}_n)}[\nabla_{\boldsymbol{\theta}_0}p(\boldsymbol{\theta}_0)r(\tau)]$$
$$= \int p(\cdot|\boldsymbol{\theta}_n)\nabla_{\boldsymbol{\theta}_0}p(\boldsymbol{\theta}_0)r(\tau)d\tau$$
$$= \int \frac{p(\cdot|\boldsymbol{\theta}_0)}{p(\cdot|\boldsymbol{\theta}_0)}p(\cdot|\boldsymbol{\theta}_n)\nabla_{\boldsymbol{\theta}_0}p(\boldsymbol{\theta}_0)r(\tau)d\tau$$
$$= \int p(\cdot|\boldsymbol{\theta}_0)\frac{p(\cdot|\boldsymbol{\theta}_n)}{p(\cdot|\boldsymbol{\theta}_0)}\nabla_{\boldsymbol{\theta}_0}p(\boldsymbol{\theta}_0)r(\tau)d\tau$$
$$= \mathbb{E}_{\tau \sim p(\cdot|\boldsymbol{\theta}_0)}\left[\frac{p(\cdot|\boldsymbol{\theta}_n)}{p(\cdot|\boldsymbol{\theta}_0)}\nabla_{\boldsymbol{\theta}_0}p(\boldsymbol{\theta}_0)r(\tau)\right]$$
$$= \frac{p(\cdot|\boldsymbol{\theta}_n)}{p(\cdot|\boldsymbol{\theta}_0)}g(\tau_0|\boldsymbol{\theta}_0)$$

Then,

$$\omega(\tau|\boldsymbol{\theta}_n, \boldsymbol{\theta}_0)g(\tau_n|\boldsymbol{\theta}_0) = \frac{p(\cdot|\boldsymbol{\theta}_0)}{p(\cdot|\boldsymbol{\theta}_n)}g(\tau_n|\boldsymbol{\theta}_0)$$
$$= g(\tau_0|\boldsymbol{\theta}_0)$$

which gives the lemma. $\square$

**Lemma 9** (Adapted from [19]). *Let $\omega(\tau|\boldsymbol{\theta}_1, \boldsymbol{\theta}_2) = p(\tau|\boldsymbol{\theta}_1)/p(\tau|\boldsymbol{\theta}_2)$, under Assumptions 3 and 5, it holds that*

$$Var(\omega(\tau|\boldsymbol{\theta}_1, \boldsymbol{\theta}_2)) \leq C_w\|\boldsymbol{\theta}_1 - \boldsymbol{\theta}_2\|^2$$

*where $C_w = H(2HG^2 + M)(W + 1)$. Furthermore, we have*

$$\mathbb{E}_{\tau_n^t}\|1 - \omega(\tau_n^t|\boldsymbol{\theta}_n^t, \boldsymbol{\theta}_0^t)\|^2$$
$$= Var_{\boldsymbol{\theta}_n^t, \boldsymbol{\theta}_0^t}(\omega(\tau_n^t|\boldsymbol{\theta}_n^t, \boldsymbol{\theta}_0^t))$$
$$\leq C_w\|\boldsymbol{\theta}_n^t - \boldsymbol{\theta}_0^t\|^2$$

*Proof.* The proof can be found in Xu et al. [19]. $\square$

**Lemma 10.** *For $X_1, X_2 \in \mathbb{R}^d$, we have*
$$\|X_1 + X_2\|^2 \leq 2\|X_1\|^2 + 2\|X_2\|^2$$

**Lemma 11.**
$$\mathbb{E}_{\tau_n^t}[\|v_n^t\|^2] \leq (2L_g + 2C_g^2 C_w)\left\|\boldsymbol{\theta}_n^t - \boldsymbol{\theta}_0^t\right\|^2 + 2\left\|\nabla J(\boldsymbol{\theta}_n^t)\right\|^2 + 2\left\|e_t\right\|^2$$

*Proof.* We follow the suggestion of Lei et al. [35] to set $b_t = 1$ to deliver better theoretical results. However in our experiments, we do allow $b_t$ to be sampled from different values. With $b_t = 1$ and $\mu_t = \frac{1}{|\mathcal{G}_t|}\sum_{k \in \mathcal{G}_t}\mu_t^{(k)}$, we have the flowing definition according to Algorithm 1:

$$v_n^t \triangleq g(\tau_n^t \mid \boldsymbol{\theta}_n^t) - \omega(\tau_n^t \mid \boldsymbol{\theta}_n^t, \boldsymbol{\theta}_0^t)g(\tau_n^t \mid \boldsymbol{\theta}_0^t) + u_t \tag{11-12}$$

which is the SCSG update step. Define $e_t \triangleq u_t - \nabla J(\boldsymbol{\theta}_0^t)$, we then have

$$\mathbb{E}_{\tau_n^t}[v_n^t] = \nabla J(\boldsymbol{\theta}_n^t) - \nabla J(\boldsymbol{\theta}_0^t) + e_t + \nabla J(\boldsymbol{\theta}_0^t)$$
$$= \nabla J(\boldsymbol{\theta}_n^t) + e_t \tag{11-13}$$

Note that $\nabla J(\boldsymbol{\theta}_n^t) - \nabla J(\boldsymbol{\theta}_0^t) = \mathbb{E}_{\tau_n^t}[g(\tau_n^t \mid \boldsymbol{\theta}_n^t) - \omega(\tau_n^t \mid \boldsymbol{\theta}_n^t, \boldsymbol{\theta}_0^t)g(\tau_n^t \mid \boldsymbol{\theta}_0^t)]$ as we have showed that the importance weighting term results in unbiased estimation of the true gradient in Lemma 8. Then from $\mathbb{E}\|X\|^2 = \mathbb{E}\|X - \mathbb{E}X\|^2 + \|\mathbb{E}X\|^2$,

$$\mathbb{E}_{\tau_n^t}[\|v_n^t\|^2] = \mathbb{E}_{\tau_n^t}\left\|v_n^t - \mathbb{E}_{\tau_n^t}[v_n^t]\right\|^2 + \left\|\mathbb{E}_{\tau_n^t}[v_n^t]\right\|^2$$

$$= \mathbb{E}_{\tau_n^t}\left\|g(\tau_n^t \mid \boldsymbol{\theta}_n^t) - \omega(\tau_n^t \mid \boldsymbol{\theta}_n^t, \boldsymbol{\theta}_0^t)g(\tau_n^t \mid \boldsymbol{\theta}_0^t) + u_t - (\nabla J(\boldsymbol{\theta}_n^t) + e_t)\right\|^2 + \left\|\mathbb{E}_{\tau_n^t}[v_n^t]\right\|^2$$

$$= \mathbb{E}_{\tau_n^t}\left\|g(\tau_n^t \mid \boldsymbol{\theta}_n^t) - \omega(\tau_n^t \mid \boldsymbol{\theta}_n^t, \boldsymbol{\theta}_0^t)g(\tau_n^t \mid \boldsymbol{\theta}_0^t) - (\nabla J(\boldsymbol{\theta}_n^t) - \nabla J(\boldsymbol{\theta}_0^t))\right\|^2 + \left\|\nabla J(\boldsymbol{\theta}_n^t) + e_t\right\|^2$$

$$\leq \mathbb{E}_{\tau_n^t}\left\|g(\tau_n^t \mid \boldsymbol{\theta}_n^t) - \omega(\tau_n^t \mid \boldsymbol{\theta}_n^t, \boldsymbol{\theta}_0^t)g(\tau_n^t \mid \boldsymbol{\theta}_0^t)\right\|^2 + 2\left\|\nabla J(\boldsymbol{\theta}_n^t)\right\|^2 + 2\left\|e_t\right\|^2 \tag{11-14}$$

where (11-14) follows from $\mathbb{E}\|X - \mathbb{E}X\|^2 \leq \mathbb{E}\|X\|^2$ and Lemma 10. Note that

$$\mathbb{E}_{\tau_n^t}\left\|g(\tau_n^t \mid \boldsymbol{\theta}_n^t) - \omega(\tau_n^t \mid \boldsymbol{\theta}_n^t, \boldsymbol{\theta}_0^t)g(\tau_n^t \mid \boldsymbol{\theta}_0^t)\right\|^2$$

$$= \mathbb{E}_{\tau_n^t}\|g(\tau_n^t \mid \boldsymbol{\theta}_n^t) + g(\tau_n^t \mid \boldsymbol{\theta}_0^t) - g(\tau_n^t \mid \boldsymbol{\theta}_0^t) - \omega(\tau_n^t \mid \boldsymbol{\theta}_n^t, \boldsymbol{\theta}_0^t)g(\tau_n^t \mid \boldsymbol{\theta}_0^t)\|^2$$

$$= \mathbb{E}_{\tau_n^t}\|g(\tau_n^t \mid \boldsymbol{\theta}_n^t) - g(\tau_n^t \mid \boldsymbol{\theta}_0^t) + (1 - \omega(\tau_n^t \mid \boldsymbol{\theta}_n^t, \boldsymbol{\theta}_0^t))g(\tau_n^t \mid \boldsymbol{\theta}_0^t)\|^2$$

$$\leq 2\mathbb{E}_{\tau_n^t}\left\|g(\tau_n^t \mid \boldsymbol{\theta}_n^t) - g(\tau_n^t \mid \boldsymbol{\theta}_0^t)\right\|^2 + 2\mathbb{E}_{\tau_n^t}\left\|(1 - \omega(\tau_n^t \mid \boldsymbol{\theta}_n^t, \boldsymbol{\theta}_0^t))g(\tau_n^t \mid \boldsymbol{\theta}_0^t)\right\|^2 \tag{11-15}$$

where (11-15) follows from Lemma 10. Combining (11-14) and (11-15), we have

$$\mathbb{E}_{\tau_n^t}[\|v_n^t\|^2] \leq 2\mathbb{E}_{\tau_n^t}\left\|g(\tau_n^t \mid \boldsymbol{\theta}_n^t) - g(\tau_n^t \mid \boldsymbol{\theta}_0^t)\right\|^2$$

$$+ 2\mathbb{E}_{\tau_n^t}\left\|(1 - \omega(\tau_n^t|\boldsymbol{\theta}_n^t, \boldsymbol{\theta}_0^t))g(\tau_n^t \mid \boldsymbol{\theta}_0^t)\right\|^2 + 2\left\|\nabla J(\boldsymbol{\theta}_n^t)\right\|^2 + 2\left\|e_t\right\|^2$$

$$\leq 2L_g\left\|\boldsymbol{\theta}_n^t - \boldsymbol{\theta}_0^t\right\|^2 + 2C_g^2\mathbb{E}_{\tau_n^t}\|(1 - \omega(\tau_n^t|\boldsymbol{\theta}_n^t, \boldsymbol{\theta}_0^t))\|^2 + 2\left\|\nabla J(\boldsymbol{\theta}_n^t)\right\|^2 + 2\left\|e_t\right\|^2 \tag{11-16}$$

$$\leq 2L_g\left\|\boldsymbol{\theta}_n^t - \boldsymbol{\theta}_0^t\right\|^2 + 2C_g^2 C_w\left\|\boldsymbol{\theta}_n^t - \boldsymbol{\theta}_0^t\right\|^2 + 2\left\|\nabla J(\boldsymbol{\theta}_n^t)\right\|^2 + 2\|e_t\|^2 \tag{11-17}$$

$$= (2L_g + 2C_g^2 C_w)\left\|\boldsymbol{\theta}_n^t - \boldsymbol{\theta}_0^t\right\|^2 + 2\left\|\nabla J(\boldsymbol{\theta}_n^t)\right\|^2 + 2\left\|e_t\right\|^2 \tag{11-18}$$

where (11-16) is from Lemma 4 and (11-17) follows from Lemma 9 $\qquad\square$

**Lemma 12.**

$$\eta_t\mathbb{E}\left\langle e_t, \mathbb{E}\nabla J(\tilde{\boldsymbol{\theta}}_t)\right\rangle = \frac{1}{B_t}\mathbb{E}\left\langle e_t, \tilde{\boldsymbol{\theta}}_t - \tilde{\boldsymbol{\theta}}_{t-1}\right\rangle - \eta_t\mathbb{E}\left\|e_t\right\|^2$$

*Proof.* Consider $M_n^t = \langle e_t, \boldsymbol{\theta}_n^t - \boldsymbol{\theta}_0^t\rangle$. We have
$$M_{n+1}^t - M_n^t = \langle e_t, \boldsymbol{\theta}_{n+1}^t - \boldsymbol{\theta}_n^t\rangle = \eta_t\langle e_t, v_n^t\rangle$$

Taking expectation with respect to $\tau_n^t$, we have
$$\mathbb{E}_{\tau_n^t}\left[M_{n+1}^t - M_n^t\right] = \eta_t\left\langle e_t, \mathbb{E}_{\tau_n^t}[v_n^t]\right\rangle$$
$$= \eta_t\left\langle e_t, \nabla J(\boldsymbol{\theta}_n^t)\right\rangle + \eta_t\left\|e_t\right\|^2$$

following from (11-13). Use $\mathbb{E}_t$ to denote the expectation with respect to all trajectories $\{\tau_1^t, \tau_2^t, ...\}$, given $N_t$. Since $\{\tau_1^t, \tau_2^t, ...\}$ are independent of $N_t$, $\mathbb{E}_t$ is equivalently the expectation with respect to $\{\tau_1^t, \tau_2^t, ...\}$. We have

$$\mathbb{E}_t[M_{n+1}^t - M_n^t] = \eta_t\left\langle e_t, \mathbb{E}_t\nabla J(\boldsymbol{\theta}_n^t)\right\rangle + \eta_t\left\|e_t\right\|^2$$

Taking $n = N_t$ and denoting $\mathbb{E}_{N_t}$ the expectation w.r.t. $N_t$, we have

$$\mathbb{E}_{N_t}\mathbb{E}_t(M_{N_t+1}^t - M_{N_t}^t) = \eta_t\langle e_t, \mathbb{E}_{N_t}\mathbb{E}_t\nabla J(\boldsymbol{\theta}_{N_t}^t)\rangle + \eta_t\left\|e_t\right\|^2.$$

Using Fubini's theorem, Lemma 16 and using the fact $\boldsymbol{\theta}_{N_t}^t = \tilde{\boldsymbol{\theta}}_t$ and $\boldsymbol{\theta}_0^t = \tilde{\boldsymbol{\theta}}_{t-1}$,

$$\mathbb{E}_{N_t}\mathbb{E}_t(M_{N_t+1}^t - M_{N_t}^t) = -\mathbb{E}_t\mathbb{E}_{N_t}(M_{N_t}^t - M_{N_t+1}^t)$$

$$= -(\frac{1}{B_t/(B_t+1)} - 1)(M_0^t - \mathbb{E}_{N_t}\mathbb{E}_t M_{N_t}^t)$$

$$= \frac{1}{B_t}\mathbb{E}_{N_t}\mathbb{E}_t\left\langle e_t, \tilde{\boldsymbol{\theta}}_t - \tilde{\boldsymbol{\theta}}_{t-1}\right\rangle$$

$$= \eta_t\left\langle e_t, \mathbb{E}_{N_t}\mathbb{E}_t\nabla J(\boldsymbol{\theta}_{N_t}^t)\right\rangle + \eta_t\left\|e_t\right\|^2$$

Taking expectation with respect to the whole past yields the lemma. $\qquad\square$

**Lemma 13.**

$$-2\eta_t\mathbb{E}\langle e_t, \tilde{\boldsymbol{\theta}}_t - \tilde{\boldsymbol{\theta}}_{t-1}\rangle \leq \left[-\frac{1}{B_t} + \eta_t^2(2L_g + 2C_g^2C_w)\right]\mathbb{E}\|\tilde{\boldsymbol{\theta}}_t - \tilde{\boldsymbol{\theta}}_{t-1}\|^2 + 2\eta_t^2\mathbb{E}\|e_t\|^2$$
$$+2\eta_t\mathbb{E}\langle\nabla J(\tilde{\boldsymbol{\theta}}_t), \tilde{\boldsymbol{\theta}}_t - \tilde{\boldsymbol{\theta}}_{t-1}\rangle + 2\eta_t^2\mathbb{E}\|\nabla J(\tilde{\boldsymbol{\theta}}_t)\|^2$$

*Proof.* We have from the update equation $\boldsymbol{\theta}_{n+1}^t = \boldsymbol{\theta}_n^t + \eta_t v_n^t$, then,

$$\mathbb{E}_{\tau_n^t}\|\boldsymbol{\theta}_{n+1}^t - \boldsymbol{\theta}_0^t\|^2 = \mathbb{E}_{\tau_n^t}\|\boldsymbol{\theta}_n^t + \eta_t v_n^t - \boldsymbol{\theta}_0^t\|^2$$
$$= \|\boldsymbol{\theta}_n^t - \boldsymbol{\theta}_0^t\|^2 + \eta_t^2\mathbb{E}_{\tau_n^t}\|v_n^t\|^2 + 2\eta_t\langle\mathbb{E}_{\tau_n^t}[v_n^t], \boldsymbol{\theta}_n^t - \boldsymbol{\theta}_0^t\rangle$$
$$\leq \|\boldsymbol{\theta}_n^t - \boldsymbol{\theta}_0^t\|^2 + \eta_t^2[(2L_g + 2C_g^2C_w)\|\boldsymbol{\theta}_n^t - \boldsymbol{\theta}_0^t\|^2 + 2\|\nabla J(\boldsymbol{\theta}_n^t)\|^2 + 2\|e_t\|^2]$$
$$+ 2\eta_t\langle e_t, \boldsymbol{\theta}_n^t - \boldsymbol{\theta}_0^t\rangle + 2\eta_t\langle\nabla J(\boldsymbol{\theta}_n^t), \boldsymbol{\theta}_n^t - \boldsymbol{\theta}_0^t\rangle \qquad (13\text{-}19)$$
$$= [1 + \eta_t^2(2L_g + 2C_g^2C_w)]\|\boldsymbol{\theta}_n^t - \boldsymbol{\theta}_0^t\|^2 + 2\eta_t\langle\nabla J(\boldsymbol{\theta}_n^t), \boldsymbol{\theta}_n^t - \boldsymbol{\theta}_0^t\rangle$$
$$+ 2\eta_t\langle e_t, \boldsymbol{\theta}_n^t - \boldsymbol{\theta}_0^t\rangle + 2\eta_t^2\|\nabla J(\boldsymbol{\theta}_n^t)\|^2 + 2\eta_t^2\|e_t\|^2$$

where (13-19) follows the result of (11-18). Use $\mathbb{E}_t$ to denote the expectation with respect to all trajectories $\{\tau_1^t, \tau_2^t, ...\}$, given $N_t$. Since $\{\tau_1^t, \tau_2^t, ...\}$ are independent of $N_t$, $\mathbb{E}_t$ is equivalently the expectation with respect to $\{\tau_1^t, \tau_2^t, ...\}$. We have

$$\mathbb{E}_t\|\boldsymbol{\theta}_{n+1}^t - \boldsymbol{\theta}_0^t\|^2 \leq [1 + \eta_t^2(2L_g + 2C_g^2C_w)]\mathbb{E}_t\|\boldsymbol{\theta}_n^t - \boldsymbol{\theta}_0^t\|^2 + 2\eta_t\mathbb{E}_t\langle\nabla J(\boldsymbol{\theta}_n^t), \boldsymbol{\theta}_n^t - \boldsymbol{\theta}_0^t\rangle$$
$$+ 2\eta_t\mathbb{E}_t\langle e_t, \boldsymbol{\theta}_n^t - \boldsymbol{\theta}_0^t\rangle + 2\eta_t^2\mathbb{E}_t\|\nabla J(\boldsymbol{\theta}_n^t)\|^2 + 2\eta_t^2\|e_t\|^2$$

Now taking $n = N_t$ and denoting $E_{N_t}$ the expectation w.r.t. $N_t$ we have

$$-2\eta_t\mathbb{E}_{N_t}\mathbb{E}_t\left\langle e_t, \boldsymbol{\theta}_{N_t}^t - \boldsymbol{\theta}_0^t\right\rangle$$
$$\leq [1 + \eta_t^2(2L_g + 2C_g^2C_w)]\mathbb{E}_{N_t}\mathbb{E}_t\left\|\boldsymbol{\theta}_{N_t}^t - \boldsymbol{\theta}_0^t\right\|^2 - \mathbb{E}_{N_t}\mathbb{E}_t\left\|\boldsymbol{\theta}_{N_t+1}^t - \boldsymbol{\theta}_0^t\right\|^2$$
$$+ 2\eta_t\mathbb{E}_{N_t}\mathbb{E}_t\left\langle\nabla J(\boldsymbol{\theta}_{N_t}^t), \boldsymbol{\theta}_{N_t}^t - \boldsymbol{\theta}_0^t\right\rangle + 2\eta_t^2\mathbb{E}_{N_t}\mathbb{E}_t\left\|\nabla J(\boldsymbol{\theta}_{N_t}^t)\right\|^2 + 2\eta_t^2\left\|e_t\right\|^2$$
$$= \left[-\frac{1}{B_t} + \eta_t^2(2L_g + 2C_g^2C_w)\right]\mathbb{E}_{N_t}\mathbb{E}_t\left\|\boldsymbol{\theta}_{N_t}^t - \boldsymbol{\theta}_0^t\right\|^2$$
$$+ 2\eta_t\mathbb{E}_{N_t}\mathbb{E}_t\left\langle\nabla J(\boldsymbol{\theta}_{N_t}^t), \boldsymbol{\theta}_{N_t}^t - \boldsymbol{\theta}_0^t\right\rangle + 2\eta_t^2\mathbb{E}_{N_t}\mathbb{E}_t\left\|\nabla J(\boldsymbol{\theta}_{N_t}^t)\right\|^2 + 2\eta_t^2\left\|e_t\right\|^2 \quad (13\text{-}20)$$

where (13-20) follows Lemma 16 using Fubini's theorem. Rearranging, replacing $\boldsymbol{\theta}_{N_t}^t = \tilde{\boldsymbol{\theta}}_t$ and $\boldsymbol{\theta}_0^t = \tilde{\boldsymbol{\theta}}_{t-1}$ and taking expectation w.r.t the whole past yields the lemma. $\qquad\square$

**Lemma 14** (Pinelis' inequality [76]; Lemma 2.4 [46]). *Let the sequence of random variables $X_1, X_2, ..., X_N \in \mathbb{R}^d$ represent a random process such that we have $\mathbb{E}[X_n|X_1, ..., X_{n-1}]$ and $\|X_n\| \leq M$. Then,*

$$\mathbb{P}\left[\|X_1 + ... + X_N\|^2 \leq 2\log(2/\delta)M^2N\right] \geq 1 - \delta$$

**Lemma 15** (Adapted from [48]). *If we choose $\delta$ and $B_t$ in Algorithm 1 such that:*
*(i)* $e^{\frac{\delta B_t}{2(1-2\delta)}} \leq \frac{2K}{\delta} \leq e^{\frac{B_t}{2}}$
*(ii)* $\delta \leq \frac{1}{5KB_t}$
*then we have the following bound for $\mathbb{E}\|e_t\|^2$:*

$$\mathbb{E}\|e_t\|^2 \leq \frac{4\sigma^2}{(1-\alpha)^2KB_t} + \frac{48\alpha^2\sigma^2V}{(1-\alpha)^2B_t}$$

*Proof.* The proof of this lemma is similar to that of Lemma 7 of Khanduri et al. [48]. The key difference lays on the base conditions used to define the probabilistic events.

In FedPG-BR, the following refined conditions (results of Claims D.1 and D.2) are used,

$$\|\mu_t^{\text{mom}} - \nabla J(\boldsymbol{\theta})\| \leq \sigma, \|\mu_t^{(k)} - \mu_t^{\text{mom}}\| \leq 2\sigma, \|\mu_t^{(k)} - \nabla J(\boldsymbol{\theta})\| \leq 3\sigma, \forall k \in \mathcal{G}_t$$

whereas Khanduri et al. [48] needs the following:

$$\|\mu_t^{\mathrm{med}} - \nabla J(\boldsymbol{\theta})\| \le 3\sigma, \|\mu_t^{(k)} - \mu_t^{\mathrm{med}}\| \le 4\sigma, \|\mu_t^{(k)} - \nabla J(\boldsymbol{\theta})\| \le 7\sigma, \forall k \in \mathcal{G}_t$$

The detailed proof of Lemma 15 can be obtained following the derivation of Lemma 7 of Khanduri et al. [48] by modifying the base conditions. □

**Lemma 16.** *If* $N \sim Geom(\Gamma)$ *for* $\Gamma > 0$. *Then for any sequence* $D_0, D_1, ...$ *with* $\mathbb{E}\|D_N\| \le \infty$, *we have*

$$\mathbb{E}\left[D_N - D_{N+1}\right] = (\frac{1}{\Gamma} - 1)(D_0 - \mathbb{E}D_N)$$

*Proof.* The proof can be found in Lei et al. [35]. □

**Lemma 17** (Young's inequality (Peter-Paul inequality))**.** *For all real numbers* $a$ *and* $b$ *and all* $\beta > 0$, *we have*

$$ab \le \frac{a^2}{2\beta} + \frac{\beta b^2}{2}$$

# F    Experimental details

## F.1    Hyperparameters

We follow the setups of SVRPG [18] to parameterize the policies using neural networks. For all the algorithms under comparison in the experiments (Section 5), Adam[77] is used as the gradient optimizer. The 10 random seeds are $[0-9]$. All other hyperparameters used in all the experiments are reported in Table 2.

Table 2: Hyperparameters used in the experiments.

| Hyperparameters | Algorithms | CartPole-v1 | LunarLander-v2 | HalfCheetach-v2 |
|---|---|---|---|---|
| NN policy | - | Categorical MLP | Categorical MLP | Gaussian MLP |
| NN hidden weights | - | 16,16 | 64,64 | 64,64 |
| NN activation | - | ReLU | Tanh | Tanh |
| NN output activation | - | Tanh | Tanh | Tanh |
| Step size (Adam) $\eta$ | - | 1e-3 | 1e-3 | 8e-5 |
| Discount factor $\gamma$ | - | 0.999 | 0.990 | 0.995 |
| Maximum trajectories | - | 5000 | 10000 | 10000 |
| Task horizon $H$ (for training) | - | 500 | 1000 | 500 |
| Task horizon $H$ (for test) | - | 500 | 1000 | 1000 |
| $\alpha$ (for practical setup) | - | 0.3 | 0.3 | 0.3 |
| Number of runs | - | 10 | 10 | 10 |
| | GPOMDP | 16 | 32 | 48 |
| Batch size $B_t$ | SVRPG | 16 | 32 | 48 |
| | FedPG-BR | sampled from $[12, 20]$ | sampled from $[26, 38]$ | sampled from $[46, 50]$ |
| | GPOMDP | - | - | - |
| Mini-Batch size $b_t$ | SVRPG | 4 | 8 | 16 |
| | FedPG-BR | 4 | 8 | 16 |
| | GPOMDP | 1 | 1 | 1 |
| Number of steps $N_t$ | SVRPG | 3 | 3 | 3 |
| | FedPG-BR | $N_t \sim Geom(\frac{B_t}{B_t+b_t})$ | $N_t \sim Geom(\frac{B_t}{B_t+b_t})$ | $N_t \sim Geom(\frac{B_t}{B_t+b_t})$ |
| Variance bound $\sigma$ | GPOMDP | - | - | - |
| (Estimated by server) | SVRPG | - | - | - |
| | FedPG-BR | 0.06 | 0.07 | 0.9 |
| | GPOMDP | - | - | - |
| Confidence parameter $\delta$ | SVRPG | - | - | - |
| | FedPG-BR | 0.6 | 0.6 | 0.6 |

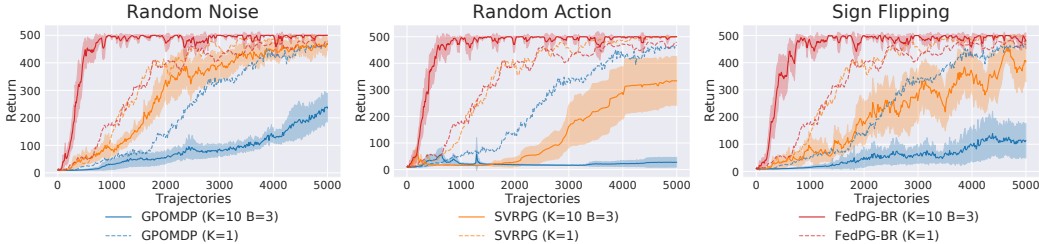

Figure 5: Performance of FedPG-BR in practical systems with $\alpha > 0$ for CartPole. Each subplot corresponds to a different type of Byzantine failure exercised by the 3 Byzantine agents.

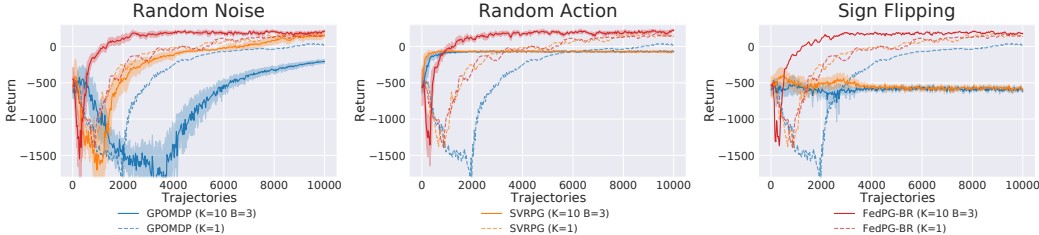

Figure 6: Performance of FedPG-BR in practical systems with $\alpha > 0$ for LunarLander. Each subplot corresponds to a different type of Byzantine failure exercised by the 3 Byzantine agents.

## F.2 Computing Infrastructure

All experiments are conducted on a computing server without GPUs. The server is equipped with 14 cores (28 threads) *Intel(R) Core(TM) i9-10940X CPU @ 3.30GHz* and 64G memory. The average runtime for each run of FedPG-BR (K=10 B=3) is 2.5 hours for the CartPole task, 4 hours for the HalfCheetah task, and 12 hours for the LunarLander task.

## G  Additional experiments

### G.1  Performance of FedPG-BR in practical systems with $\alpha > 0$ for the CartPole and the LunarLander tasks

The results for the CartPole and the LunarLander tasks which yield the same insights as discussed in experiments (Section 5) are plotted in Figure 5 and Figure 6. As discussed earlier, for both GPOMDP and SVRPG, the federation of more agents in practical systems which are subject to the presence of Byzantine agents, i.e., random failures or adversarial attacks, causes the performance of their federation to be worse than that in the single-agent setting. In particular, RA agents (middle figure) and SF agents (right figure) render GPOMDP and SVRPG unlearnable, i.e., unable to converge at all. This is in contrast to the performance of FedPG-BR. That is, FedPG-BR ($K = 10B = 3$) is able to deliver superior performances even in the presence of Byzantine agents for all three tasks: CartPole (Figure 5), LunarLander (Figure 6), and HalfCheetah (Figure 2 in Section 5). This provides an assurance on the reliability of our FedPG-BR algorithm to promote its practical deployment, and significantly improves the practicality of FRL.

### G.2  Performance of FedPG-BR against the Variance Attack

We have discussed in Section 3.2 where the high variance in PG estimation renders the FRL system vulnlerable to variance-based attacks such as the Variance Attack (VA) proposed by Baruch et al. [47]. The VA attackers collude together to estimate the population mean and the standard-deviation of gradients at each round, and move the mean by the largest value such that their values are still within the population variance. Intuitively, this non-omniscient attack works by exploiting the high variance in gradient estimation of the population and crafting values that contribute most to the population variance, hence gradually shifting the population mean. According to Cao et al. [20],

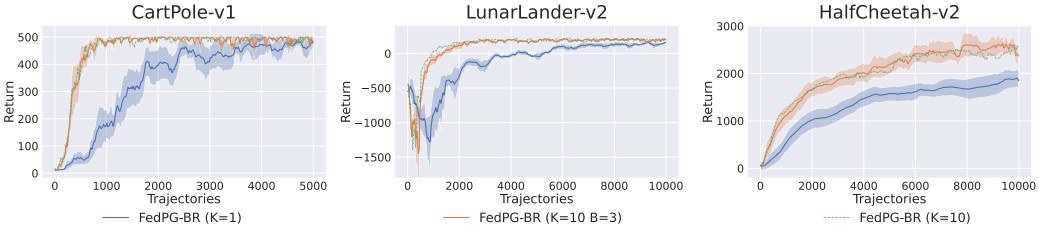

Figure 7: Performance of FedPG-BR in practical systems with $\alpha > 0$ for CartPole. Among the $K = 10$ participating agents, 3 Byzantine agents are colluding together to launch the VA attack.

existing defenses will fail to remove those non-omniscient attackers and the convergence will be significantly worsened if the population variance is large enough.

We are thus motivated to look for solutions that theoretically reduce the variance in policy gradient estimation. Inspired by the variance-reduced policy gradient works [e.g., 18, 19], we adapt the SCSG optimization [35] to our federated policy gradient framework for a refined control over the estimation variance. Through our adaptation, we are able to control the variance by the semi-stochastic gradient (line 11 in Algorithm 1), hence resulting in the fault-tolerant FRL system that can defend the VA attackers. Each plot in Figure 7 shows the experiment for each of the three tasks correspondingly, where 3 Byzantine agents are implemented as the VA attackers [20] ($z^{max}$ is 0.18 in our setup). We again include the corresponding single-agent performance ($K = 1$) and the federation of 10 good agents ($K = 10$) in the plots for reference. The results show that in all three tasks, FedPG-BR ($K = 10 B = 3$) still manages to significantly outperform FedPG-BR ($K = 1$) in the single-agent setting. Furthermore, the performance of FedPG-BR ($K = 10 B = 3$) is barely worsened compared with FedPG-BR ($K = 10$) with 10 good agents. This shows that, with the adaptation of SCSG, our fault-tolerant FRL system can perfectly defend the VA attack from the literature, which further corroborates our analysis on our Byzantine filtering step (Section 3.3) showing that if gradients from Byzantine agents are not filtered out, their impact is limited since their maximum distance to $\nabla J(\boldsymbol{\theta}_0^t)$ is bounded by $3\sigma$ (Claim D.2).