# OpenReview forum: "Fault-Tolerant Federated Reinforcement Learning with Theoretical Guarantee"
_NeurIPS.cc/2021/Conference — NeurIPS 2021 Poster_

### Official Review · Reviewer_pSwe · 2021-07-16

**Rating:** 6
**Confidence:** 3

**Summary:**

This paper proposes a federated reinforcement learning (FRL) algorithm that uses a variance-reduced federated policy gradient framework. The proposed framework is proven to have convergence and robustness guatantees (can tolerant the failures of up to half of the agents), with some assumptions on the environment and the agents. Both theoretical results and empirical results justify that the proposed federation can boost the sample efficiency of RL agents.

**Limitations And Societal Impact:**

The experiment environments used in this paper is relatively simple, so it is not clear how the proposed algorithm scales to large environments and a large number of agents.

**Main Review:**

This paper proposes the first FRL method with formal convergence and robustness guarantees, which I think is a significant contribution to the community. The theoretical analysis seems to be correct, and the experiment results do support the theory. This paper is in generally well-written and easy to follow. I am not very familar with the related work, so it is hard to justify the novelty of the proposed idea. But this paper provides explanations about technical challenges that can distinguish this paper from prior works.

My main concern is that, although the analysis makes intuitive sense, is mainly built on some assumptions which might not be realistic. For example, assumption 2 is important to the algorithm and the guarantees, and the variance bound $\sigma$ is also used as a hyperparameter in Algorithm 1.1. How can we know the value of $\sigma$ in practice? How possible is that $\sigma$ exists and is not too large? If I am understanding it correctly, the gradient estimator lives in a high-dimensional space, and this assumption should hold for any trajectory generated from the environment given a certain policy, so $\sigma$ can be very large, which also makes the sample complexity high.


**Time Spent Reviewing:**

3

---

> ### Author Response · Authors · 2021-08-10
> **Response to Reviewer pSwe**
>
> Thanks for taking the time to review our paper and offer valuable comments. We appreciate that the reviewer finds our contribution significant. We hope that the following response will help clear the concerns.
>
> - The value of $\sigma$ is the maximum difference between optimal gradient $\nabla J(\theta)$ and the gradient estimate $g(\tau | \theta)$ w.r.t. any trajectories induced by policy $\pi_{\theta}$. For many complex real-world problems with continuous, high-dimensional controls, $\sigma$ may be upper-bounded, provided that the MDP is Lipschitz continuous. The deviation can be obtained by referring to Proposition 2 of [1].
>
> - Assumption 2 can be relaxed to $\mathbb{E} \|| g(\tau | \theta) - \nabla J(\theta) \|| \leq \theta$ which is a standard assumption commonly used in stochastic non-convex optimization [2-3, etc.]
>
> > **How can we know the value of $\sigma$ in practice?**
>
> In practice, we estimate $\sigma$ by using the server to sample a few trajectories (as stated in Table 2 in Appendix F), prior to the federation.
>
> > **How possible is that $\sigma$ exists and is not too large?**
>
> As explained above, $\sigma$ is the difference in gradients which always exists and its magnitude is commonly seen to be not significant in the analysis of stochastic optimization. We reported the value of $\sigma$ in Table 2 in Appendix F. The values of $\sigma$ we estimated for the three tasks are 0.06, 0.07, 0.9.
>
> > **…, so $\sigma$ can be very large, which also makes the sample complexity high.**
>
> Similarly, the value of $\sigma$ is not significant to the sample complexity bound as it is only a constant whose magnitude is limited.
>
> **Limitations And Social Impacts.** We will experiment with more complex environments and a larger number of agents in future work. We can expect similar results with a larger number of agents, as we have both theoretically and empirically verified.  To the best of our knowledge, most current works in multi-agents / federated RL are mainly experimenting with mujoco environments with 3-10 agents [4-5, etc.].
>
> We hope the above clarifies and will be grateful if you can improve your score.
>
> ---
>
> [1] Pirotta, M., Restelli, M., & Bascetta, L. (2015). Policy gradient in lipschitz markov decision processes. Machine Learning, 100(2), 255-283.
>
> [2] Zeyuan Allen-Zhu and Elad Hazan. Variance reduction for faster non-convex optimization. In International conference on machine learning, pages 699–707, 2016
>
> [3] Lihua Lei, Cheng Ju, Jianbo Chen, and Michael I Jordan. Non-convex finite-sum optimization via scsg methods. In Advances in Neural Information Processing Systems, pages 2348–2358, 2017.
>
> [4] Xinle Liang, Yang Liu, Tianjian Chen, Ming Liu, and Qiang Yang. Federated transfer reinforcement learning for autonomous driving. arXiv:1910.06001, 2019.
>
> [5] Chetan Nadiger, Anil Kumar, and Sherine Abdelhak. Federated reinforcement learning for fast personalization. In 2019 IEEE Second International Conference on Artificial Intelligence and Knowledge Engineering (AIKE), pages 123–127. IEEE, 2019.

---

### Official Review · Reviewer_Ap1M · 2021-07-16

**Rating:** 6
**Confidence:** 4

**Summary:**

This paper aims to propose an algorithm for federated reinforcement learning with fault tolerance against Byzantine failure. The paper mainly considers the policy gradient method and proposes a server aggregation method with a Byzantine filter to reduce the effect of gradients from malicious agents. The paper also provides a theoretical analysis to support the design decision of the filter.

**Limitations And Societal Impact:**

See the main review for the suggestion for limitations.

**Main Review:**

It’s nice to see a fault-tolerant algorithm for FRL that is based on a theoretical guarantee. The algorithm can support up to (less than) 50% of agents being malicious which should be sufficient for real-world use cases. The empirical demonstrates the sample complexity does not degrade too much even some agents are malicious.

Although this seems to be working well. I question whether the problem setting is realistic in the real world. All the agents are assumed to have identical MDP and transition probability, which seems to be very similar to a distributed RL training setting. If the environments are all the same among all the agents, why do we need to do federated learning, we can just train everything in a centralized fashion. Federated RL often contains some kind of data heterogeneity, where either the transition probability is different or the MDP differs slightly. Would the algorithm still work well if the agents are heterogeneous (e.g. have different transition functions)? Maybe, in this case, some of the good gradients would be quite different gradients from other agents. Can you provide more insights here?

In the experiment section, you selected three attack methods: random noise, random action, flip gradient. Although these methods seem to be valid, more sophisticated attacks should be considered to support the effectiveness of the algorithm. Some examples are: [1][2]. These attack may not requires large weight variance but be able to find specific values that would degrade the performance of the RL policy. This would even worst if the malicious agent collude when optimizing for their adversarial weights.

The aggregation method seems to require a pairwise computation among the weights of all the agents which may result in O(n^2) compared to O(n) in FedAVG. This may limits the scalability of the system. Is this true? Or is there ways to mitigate this?

[1] Fang, Minghong, et al. “Local Model Poisoning Attacks to Byzantine-Robust Federated Learning”
[2] Bhagoji, Arjun Nitin, et al. "Analyzing federated learning through an adversarial lens"
[3] Bagdasaryan, Eugene, et al. "How to backdoor federated learning"



**Time Spent Reviewing:**

2

---

> ### Author Response · Authors · 2021-08-10
> **Response to Reviewer Ap1M**
>
> We thank you for taking the time to review our manuscript and we are excited to see that you like the idea of our work. We would like to start with the motivation of Federated RL and its difference to distributed RL. As the example of clinical decision support given in our introduction (lines 16-25), Federated RL enables multiple agents (hospitals), who are naturally distributed, to collaborate together to find a better medical protocol, without sharing their raw data such as patient records. Federated RL is a specialisation of distributed RL where the privacy of agents becomes a concern. The response to your specific comments is as follows.
>
> > **…why do we need to do federated learning, we can just train everything in a centralized fashion.**
>
> Centralized training of RL requires raw data of each participating agent to be collected. However, many real-world problems such as the clinical example given above do not allow us to collect raw data for such centralized training, due to the concern of privacy.
>
> > **Federated RL often contains some kind of data heterogeneity, where either the transition probability is different or the MDP differs slightly.**
>
> Thanks for the suggestion on exploring the possibilities of data heterogeneity in FRL. However, we are not aware of any theoretical work of Federated RL that tackles data heterogeneity w.r.t. different MDPs. In the current state of our manuscript, our theoretical analysis relies on the fact that the underlying MDP is the same across agents, in order to derive the sample complexities and the fault-tolerance guarantees. However, it is indeed an interesting topic to extend our theoretical analysis to account for potentially different MDPs, which is worth exploring in future works.
>
> > **Would the algorithm still work well if the agents are heterogeneous …. Can you provide more insights here?**
>
> Our current algorithm is not expected to work if agents are heterogeneous. This is because we are considering the challenging setting of fault-tolerance and our Algorithm 1.1 relies on the homogeneity of agents to filter out those faulty agents. As we have pointed out in our manuscript (lines 364-365), we have planned to explore the opportunities in the fault-tolerant federation of different policy optimization methods, which will introduce heterogeneity of agents.
>
>
> If we do not consider fault-tolerance, i.e., replace Algo 1.1 with FedAvg and experiment on ideal systems ($\alpha = 0$), then we expect that our algorithm may still perform well in real applications with heterogeneous agents.  However, the system performance will not be guaranteed to always improve with a larger number of agents, since some agents may explore poorly and contribute misleading knowledge. As mentioned above, to the best of our knowledge, the theoretical convergence of the federation of heterogeneous RL agents has not been studied and is a promising topic which we will explore.
>
> > **...more sophisticated attacks should be considered to support the effectiveness of the algorithm.**
>
> First of all, [1-3] are Byzantine attacks working on classification tasks that have well-defined supervised loss functions. However, our work is essentially reinforcement learning where the functions are much more complex. Hence, there is no guarantee that those works [1-3] are effective in attacking federated RL systems. With that said, we did have adapted a recent work on such Byzantine attack [4] for Federated RL and have included the experimental results in Section G.2. The results (Figure 7 in Appendix G) yield the same insights as discussed in Section 5 of our manuscript, demonstrating that our FT-FedScsPG is effective against such a sophisticated attack from the Byzantine-tolerant optimization literature.
>
> > **This would be even worse if the malicious agents collude when optimizing for their adversarial weights.**
>
> In lines 333-344, we have described a specifically designed attack, *FedScsPG Attack*, where the malicious agents are colluding to adjust their adversarial weights, given the knowledge of the system. Experimental results shown in Figure 3 suggest that our system is effective against those attackers, even if they collude. Similarly, in the sophisticated attack adapted from [4], malicious agents are also allowed to collude. Similar performance of our system can be drawn from Figure 7 (Appendix G).
>
> > **The aggregation method seems to require a pairwise computation among the weights of all the agents which may result in O(n^2) compared to O(n) in FedAVG. This may limit the scalability of the system. Is this true? Or is there a way to mitigate this?**
>
> This is not an issue as we can implement it using the Euclidean Distance Matrix Trick [5]. Please refer to line 22 in agent.py in our code for these implementation details.
>
> Thank you for taking the time to read our response and we hope the above helps clarify your questions. We will be grateful if you can revise your score.
>
>
> ---
>
> [4] Gilad Baruch, Moran Baruch, and Yoav Goldberg. A little is enough: Circumventing defenses for distributed learning. In H. Wallach, H. Larochelle, A. Beygelzimer, F. d'Alché-Buc, E. Fox, and R. Garnett, editors, Advances in Neural Information Processing Systems, volume 32. Curran Associates, Inc., 2019.
>
> [5] Albanie, S. (2019). Euclidean Distance Matrix Trick. Retrieved from Visual Geometry Group, University of Oxford.

---

> > ### Comment · Reviewer_Ap1M · 2021-08-20
> > **Reply to response**
> >
> > Thank you for the replies to the comments. The main concern I have is still whether the assumptions for the FRL in this paper are realistic. RL agents are often trained with simulation environments (including in this paper). If all the federated agents have access to the same simulation environment, there is no privacy concern since all agents have the same information. That's why I mentioned that it can be trained in a centralized fashion. On the other hand, if each federated agent has a different local simulation environment or is trained with real-world data/environments, then it would be difficult to maintain homogeneity of data. I find it difficult to come up with real-world RL scenarios with homogeneous agents but also require privacy to be preserved. If there are some scenarios that meet the assumptions of the paper, please feel free to comment, and it would be good to motivate that in the paper.

---

> > > ### Author Response · Authors · 2021-08-20
> > > **Further clarifications**
> > >
> > > Thank you for your reply. We would like to make a clarification on the definition of privacy in the context of FRL by referring to [1] in which the idea of FRL was first proposed. According to [1], privacy is considered to be preserved if the raw *trajectories* (such as data records about patients given in the examples of [1]) are not shared among agents. Same as our work, the existing works in FRL [e.g., 1-4] listed in our manuscript all focus on the setup of homogeneous agents with access to the same MDP (just for clarification, in all the aforementioned works including our manuscript, each agent is operating in a **separate** environment, although the underlying MDP is the same).
> > >
> > > > **I find it difficult to come up with real-world RL scenarios with homogeneous agents but also require privacy to be preserved. If there are some scenarios that meet the assumptions of the paper, please feel free to comment, and it would be good to motivate that in the paper.**
> > >
> > > Let us consider the real-world example of clinical decision support given in our introduction (also given in [1]). Different hospitals (the agents) aim to collaborate to find a *better* treatment protocol (the policy) for *one single* new disease (the task). There is supposed to be *only one* MDP representing the underlying pharmacodynamics, which means that the MDP is the same across all agents. However, preservation of the privacy of patients is still required, which can be achieved by not allowing the admission records of patients (the raw trajectories) to be shared with other hospitals. In addition, [2-4] are also working on real-world RL applications in the similar problem setting as our paper.
> > >
> > > ---
> > >
> > > [1] Zhuo, H. H., Feng, W., Lin, Y., Xu, Q., & Yang, Q. (2019). Federated deep reinforcement learning. arXiv preprint arXiv:1901.08277.
> > >
> > > [2] Chetan Nadiger, Anil Kumar, and Sherine Abdelhak. Federated reinforcement learning for fast personalization. In 2019 IEEE Second International Conference on Artificial Intelligence and Knowledge Engineering (AIKE), pages 123–127. IEEE, 2019.
> > >
> > > [3] Hyun-Kyo Lim, Ju-Bong Kim, Joo-Seong Heo, and Youn-Hee Han. Federated reinforcement learning for training control policies on multiple iot devices. Sensors, 20(5):1359, 2020.
> > >
> > > [4] Shuai Yu, Xu Chen, Zhi Zhou, Xiaowen Gong, and Di Wu. When deep reinforcement learning meets federated learning: Intelligent multi-timescale resource management for multi-access edge computing in 5G ultra dense network. IEEE Internet of Things Journal, 2020

---

> > > > ### Comment · Reviewer_Ap1M · 2021-08-30
> > > > **Reply**
> > > >
> > > > Thank you for the further response to my question. Overall, your answers address some of my concerns and I have adjusted my score.
> > > >
> > > > Though I think in many real-world scenarios, the agents still often act in environments where the MPD can be (slightly) different. For example, [3] assumes that each agent's physical environment can be a bit different. Since your work is focusing on the fault-tolerance aspect, it would be interesting to analyze how small changes to MDP can affect your algorithm's performance.

---

> > > > > ### Author Response · Authors · 2021-08-31
> > > > > **Reply**
> > > > >
> > > > > Thank you and we will explore the possibilities in future works.

---

### Official Review · Reviewer_CftT · 2021-07-17

**Rating:** 5
**Confidence:** 3

**Summary:**

The authors propose a federated reinforcement learning approach that combines existing stochastic variance-reduced policy gradient method with a robust aggregation method to achieve fault tolerance. Theoretically, the authors provide convergence and sample complexity results under reasonable assumptions. The proposed approach is tested empirically on various benchmark tasks.

**Limitations And Societal Impact:**

Yes.


**Main Review:**

Overall, the presentation of this work is clear and easy to read.

My main concern is regarding novelty. Without the fault-tolerance aspects, the algorithm and analysis would be very similar to that of [Xu et al 2020]. So the incorporation of fault-tolerance in terms of Algorithm 1.1 (ScsPG-Aggregate) is the only aspect that makes it "Federated". Other aspects of FL, such as heterogeneity of agents are not covered here.

Algorithm 1.1, however, seems quite similar to the works by Alistarh et al and Khanduri et al. I wonder whether the author can elaborate on the novelty introduced in this work? Perhaps there are additional challenges involved in integrating these works that I missed?



**Time Spent Reviewing:**

2

---

> ### Author Response · Authors · 2021-08-10
> **Response to Reviewer CftT**
>
> We thank the reviewer for the time reviewing our paper and the comments. We understand that the concern is regarding our novelty and hence, we would like to first clarify the significance of our contribution.
>
> - The idea of federated reinforcement learning (FRL) has great potential in improving the sample efficiency of practical RL applications. However, we are not aware of any existing work which studies the theoretical guarantees of FRL or conducts experiments in standard RL benchmarks, which motivates this work.
>
> - To have further assurance on practical FRL applications when multiple agents are involved, we consider the fault-tolerance aspects using the Byzantine-fault formalism. To the best of our knowledge, the Byzantine fault has never been studied in the context of FRL.
>
> - It is not straightforward to apply Byzantine-tolerant methods from supervised learning (where the objective functions are well-defined classification loss) to FRL (where the objective functions are much more complex than supervised classification loss), due to a number of technical challenges detailed in Section 3.2 of our manuscript.
>
> - Our work tackles the aforementioned challenges in the context of FRL and presents FT-FedScsPG with theoretical guarantees and sample complexity analysis, which are all verified on RL benchmarks.
>
> We believe that our work has the potential in advancing the field of FRL, as agreed by other reviewers. Our responses to your specific comments are as follows.
>
> > **Without the fault-tolerance aspects, the algorithm and analysis would be very similar to that of [Xu et al 2020].**
>
> Without the fault-tolerance aspects, (i.e., replace Algorithm 1.1. with FedAvg), our work becomes a study of the theoretical guarantees of federation of *multiple RL agents* running **SCSG** optimization where [Xu et al 2020] studies the sample efficiency of *single-agent RL* running **SVRG** optimization. Therefore, our work is only similar to [Xu et al 2020] when we are restricted to the special case where $K=1$, i.e., the single-agent setting. As a result, since our method can support the federation of $K>1$ agents, our work considers a more general setting than [Xu et al 2020]. Hence, [Xu et al 2020] and our paper belong to two different lines of works, even if our fault-tolerance aspects are ignored.
>
> > **So the incorporation of fault-tolerance in terms of Algorithm 1.1 (ScsPG-Aggregate) is the only aspect that makes it “Federated". Other aspects of FL, such as heterogeneity of agents are not covered here.**
>
> Thank you for the suggestion. As clarified above, to the best of our knowledge, our work is the first attempt to fault-tolerant FRL with theoretical guarantees, supported by carefully designed experiments. Therefore, the heterogeneity of agents in fault-tolerant FRL, although not covered in this work, is indeed an interesting topic, which we believe is worth exploring in future works. As mentioned in our manuscript (lines 364-365), we have planned to explore the possibility of federation of different policy optimization methods, which will introduce the heterogeneity of agents.
>
> > **Algorithm 1.1, however, seems quite similar to the works by Alistarh et al and Khanduri et al. I wonder whether the author can elaborate on the novelty introduced in this work? Perhaps there are additional challenges involved in integrating these works that I missed?**
>
> As clarified at the beginning, our work is the first to tackle the challenges (Section 3.2) in the context of FRL and presents FT-FedScsPG with theoretical guarantees and sample complex analysis, which are then verified on RL benchmarks. In addition, the works by [Alistarh et al.] and [Khanduri et al.] do not provide any experimental studies where all of our theoretical claims are supported by empirical results on standard RL benchmarks. Furthermore, our Algorithm 1.1 alone is inspired by but different from the work of [Alistarh et al.] and [Khanduri et al.]. Specifically, referring to Appendix D in the supplementary materials, we proposed a *new* filtering mechanism in Algorithm 1.1. This new filtering mechanism allows us to greatly reduce the detection region $S_2$, resulting in smaller leading constants in our bounds. The claims and proofs are also detailed in Appendix D.
>
> We will update our manuscript to highlight our novelty with respect to your comments. We hope that the above helps to address your concern. Considering that other reviewers have all agreed that our manuscript has good merits in advancing Federated RL, we believe that our work is worth sharing with the community and we will be grateful if you can revise your score.

---

> > ### Comment · Reviewer_CftT · 2021-08-30
> > **Response**
> >
> > Thank you for your response. Unfortunately I am still not convinced and I maintain my scores.

---

### Official Review · Reviewer_Y3yr · 2021-07-20

**Rating:** 6
**Confidence:** 3

**Summary:**

This paper describes a federated reinforcement learning architecture that is expected to be tolerant to adversarial attacks or faulty agents. The architecture assumes a percent of the nodes are Byzantine agents that are actively trying to fool the central server, and assesses algorithm convergence and performance. Theoretical guarantees support empirical performance evaluation on some benchmark domains.

I have read the other reviews and the author responses. I agree that its difficult to do anything about the exploration policy without having further information. Although I still think this will be a limitation in the real world, drowning out interesting or unique outputs. My score remains borderline on this one.

**Ethical Concerns:**

No major concerns

**Limitations And Societal Impact:**

The authors do not seem to discuss societal impacts. Although the paper is theoretical and restricted to benchmark, it is true that federated RL will probably be used in a server node setting with many agents. Some of whom will be diverse and behave differently. It is important to know whether the idea of variance based clustering and rejecting of Byzantine agents based on that will hold in this case. Although I do not think that this is a major issue for this paper as is since its not focused on applications  currently.

**Main Review:**

The main idea of the paper seems to be to cluster good agents and bad agents using some form of variance based clustering. This idea is pretty intuitive,  and the paper is well written. Overall, I think this paper does a good job of advancing federated RL. However, I do have some questions:

1. In the situation when the agents are exploring, that is in the early stage of learning, it is possible that some agents will find very different (anomalous) outcomes than others. Not in all cases are those outcomes bad. What happens here?
2. Similarly, what is there is diversity in the agent experience, and this leads to seemingly anomalous outcomes, which are then dominated out by the others that cluster together.

It seems that if the above considerations are true, then the federated RL agent will end up ignoring critical outliers or learning experiences. Is there a way the authors could evaluate against this? I did not see discussion about the exploration phase in the experiments.

The sample complexity guarantees look good, but again the thing that confuses me with those guarantees is what is the role of exploration policy? Obviously a bad exploration policy will destroy the guarantees. Are these upper bounds? I would have liked to see some more insights into what the theoretical results mean.



**Time Spent Reviewing:**

1

---

> ### Author Response · Authors · 2021-08-10
> **Response to Reviewer Y3yr**
>
> Thank you for the time reviewing our paper and the valuable feedback. We are gratified that you like our idea and acknowledge the contribution of our work. We understand that the questions are mainly regarding how we handle exploration policies and how it affects the sample complexity bounds, for which we respond as follows.
>
> **Q1.** In this case, those agents’ experiences will be filtered out by Algorithm 1.1, which is necessary to ensure the system's reliability. Note that those agents can still participate in the federation in future rounds, because our system is designed to filter only those gradients that are anomalous *in that particular round*.
>
> **Q2.** Similarly, any seemingly anomalous experiences will be ignored by the system, ensuring the system’s reliability.
>
> > **It seems that if the above considerations are true, then the federated RL agent will end up ignoring critical outliers or learning experiences. Is there a way the authors could evaluate against this? I did not see discussion about the exploration phase in the experiments.**
>
> Yes, given the current state of our system design, such critical outliers will be ignored, which is an unavoidable cost to upper bound the overall performance in the worst case, ensuring the system’s reliability. This design helps **guarantees** that the performance of the system is either *improved* or *not affected* when new agents are added to the system, hence leading to better overall performance, as demonstrated by our experiments.
>
> > **The sample complexity guarantees look good, but again the thing that confuses me with those guarantees is what is the role of exploration policy? Obviously, a bad exploration policy will destroy the guarantees. Are these upper bounds? I would have liked to see some more insights into what the theoretical results mean**
>
> Yes, our guarantees are the worst case upper-bounds where up to half of agents $(\alpha < 0.5)$ are adversarial attackers. As explained above, any seemingly anomalous experiences including bad exploration policies will be ignored. Our theoretical results are upper bounding the worst case performance of the system with respect to the number of agents, as our main story is to improve the sample efficiency of single-agent learning, through federation with more agents including potentially faulty agents. We can interpret our bounds as follows: with more agents added to the system ($K \uparrow$), the number of samples required by each agent is reduced as $\frac{1}{\epsilon^{5/3}K^{2/3}} \downarrow$.  If the added agents were faulty agents (or showing seemingly anomalous behaviors), then that experience will be ignored and the convergence is marginally worsened by $\frac{\alpha^{4/3}}{\epsilon^{5/3}}$ as $\alpha \uparrow$. Hence, the guarantees are not destroyed.
>
> We thank you for reading our response and hope that the above clarifies. We will greatly appreciate it if you can improve your score.

---

### Decision · Program_Chairs · 2021-09-27

**Decision:**

Accept (Poster)

**Comment:**

The reviewers in general agree that the paper considers an interesting new setting and this is one of the very first works that consider federated RL with theoretical guarantees, which make the reviewers in general lean towards an acceptance. But reviewers still have some concerns on the novelty of the proposed approach and whether or not the assumptions are realistic.